Revision of the Afro-Madagascan genus Costularia (Schoeneae, Cyperaceae): infrageneric relationships and species delimitation

http://orcid.org/0000-0003-0285-722X Larridon Isabel 1 2 i.larridon@kew.org
Rabarivola Linah 3
Xanthos Martin 1
http://orcid.org/0000-0002-0763-0780 Muasya A. Muthama 4
1 Identification and Naming, Royal Botanic Gardens, Kew , Richmond, Surrey , UK
2 Deparment of Biology, Systematic and Evolutionary Botany Lab, Ghent University , Gent , Belgium
3 Kew Madagascar Conservation Centre , Antananarivo , Madagascar
4 Department of Biological Sciences, Bolus Herbarium, University of Cape Town , Rondebosch , South Africa
Thiv Mike
Electronic publication date: 2019 Feb 27
Publication date: 2019
Volume: 7
Electronic Location ID: e6528
Received 2018 Nov 30; Accepted 2019 Jan 28
Copyright: © 2019 Larridon et al.
Copyright year: 2019
Copyright holder: Larridon et al.
License: This is an open access article distributed under the terms of the Creative Commons Attribution License, which permits unrestricted use, distribution, reproduction and adaptation in any medium and for any purpose provided that it is properly attributed. For attribution, the original author(s), title, publication source (PeerJ) and either DOI or URL of the article must be cited.
License URL: https://creativecommons.org/licenses/by/4.0/

Keywords: Africa, Conservation, Molecular phylogeny, Costularia, Species delimitation, Cyperaceae, Morphology, Madagascar

Funding: B.A. Krukoff Fund for the Study of African Botany Belgian National Focal Point to the Global Taxonomy Initiative with support of the Belgian Development Cooperation CBD/GTI-02/MLS/2014.266 Ghent University Department of Biology Royal Botanic Gardens, Kew Research Foundation–Flanders (FWO) Professor Isabel Larridon is supported by the B.A. Krukoff Fund For The Study Of African Botany. This study was supported by funding from the Belgian National Focal Point to the Global Taxonomy Initiative with support of the Belgian Development Cooperation (supporting a research stay by Rondro T. Ramananjanahary at Ghent University, grant n° CBD/GTI-02/MLS/2014.266), the Ghent University Department of Biology, and the Royal Botanic Gardens, Kew. The field expeditions were financed by international mobility grants of the Research Foundation–Flanders (FWO), and with support of the Department of Biology, Ghent University, Belgium. The funders had no role in study design, data collection and analysis, decision to publish, or preparation of the manuscript.

==============================
A recent molecular phylogenetic study revealed four distinct evolutionary lineages in the genus Costularia s.l. (Schoeneae, Cyperaceae, Poales). Two lineages are part of the Oreobolus clade of tribe Schoeneae: the first being a much-reduced genus Costularia s.s., and the second a lineage endemic to New Caledonia for which a new genus Chamaedendron was erected. The other two lineages were shown to be part of the Tricostularia clade of tribe Schoeneae. Based on morphological and molecular data, the genus Costularia is here redelimited to represent a monophyletic entity including 15 species, which is restricted in distribution to southeastern Africa (Malawi, Mozambique, South Africa, Swaziland, Zimbabwe), Madagascar, the Mascarenes (La Réunion, Mauritius), and the Seychelles (Mahé). Molecular phylogenetic data based on two nuclear markers (ETS, ITS) and a chloroplast marker (trnL-F) resolve the studied taxa as monophyletic where multiple accessions could be included (except for Costularia laxa and Costularia purpurea, which are now considered conspecific), and indicate that the genus dispersed once to Africa, twice to the Mascarenes, and once to the Seychelles. Two endemic species from Madagascar are here described and illustrated as new to science, as is one additional species endemic to La Réunion. Two taxa previously accepted as varieties of Costularia pantopoda are here recognised at species level (Costularia baronii and Costularia robusta). We provide a taxonomic revision including an identification key, species descriptions and illustrations, distribution maps and assessments of conservation status for all species.

Introduction

The genus Costularia C.B.Clarke (Cyperaceae tribe Schoeneae) was previously circumscribed as including 25 species (Govaerts et al., 2018). However, a recent molecular phylogenetic study firmly established the polyphyly of the genus as previously circumscribed (Larridon et al., 2018a), which was already hinted at in previous works (Seberg, 1986, 1988a, 1988b; Browning & Gordon-Gray, 1995; Bruhl, 1995; Zhang et al., 2004; Verboom, 2006; Viljoen et al., 2013) and supported in the most recent family-wide study (Semmouri et al., 2018). Larridon et al. (2018a) showed that Costularia s.l. included four distinct lineages: (1) Costularia s.s. (11 spp.) from Africa, Madagascar, the Mascarenes and Seychelles, (2) Chamaedendron Larridon (five spp.) from New Caledonia, (3) a group largely conforming to Costularia subgenus Lophoschoenus sensu Kükenthal (1939) (eight spp.) from New Caledonia and Malesia that is now considered to be part of a redelimited genus Tetraria, nom. cons. prop. (Larridon, Verboom & Muasya, 2017b, 2018b; Larridon et al., 2018a) and (4) the species Xyroschoenus hornei (C.B.Clarke) Larridon (basionym: Schoenus hornei C.B.Clarke, nom. cons. prop.; Larridon, Govaerts & Goetghebeur, 2017a, Larridon et al., 2018a) which is endemic to the Seychelles. Only the latter species and species of Costularia s.s. are found in Africa and/or on the islands in the Indian Ocean (Henriette et al., 2015; Larridon et al., 2018a). Three earlier publications revised species of Costularia s.s. (Chermezon, 1937; Kükenthal, 1939; Henriette et al., 2015) since Clarke (1897–1898) erected the genus based on the species Costularia natalensis C.B.Clarke, as well as a species now included in Capeobolus Browning (Costularia brevicaulis C.B.Clarke; Browning & Gordon-Gray, 1999). Table 1 gives an overview of the seven species of Costularia s.s. treated by Chermezon (1937), the nine species treated by Kükenthal (1939), and the 11 currently recognised species (Govaerts et al., 2018). Costularia s.s. as here accepted more or less equates to Costularia subgenus Costularia sensu Kükenthal (1939) (Larridon et al., 2018a). The Catalogue of the Vascular Plants of Madagascar states that there may still be a number of new Madagascan endemic species to describe (Tropicos.org, 2018). This study is part of a wider effort to revise genera of Cyperaceae from Africa and Madagascar (Bauters et al., 2018; Bauters, Larridon & Goetghebeur, Accepted; Galán Díaz et al., Accepted). In this paper, we aim to (1) redelimit the genus Costularia as a monophyletic entity, (2) test the relationships between the species and investigate species limits where possible based on molecular sequence data, and (3) place previously overlooked species in a phylogenetic context and formally describe them. A taxonomic treatment including an identification key to all species, species descriptions and illustrations, distribution maps, and assessments of conservation status are provided.

Table 1 Overview of the taxa of Costularia s.s. as accepted in literature.

Chermezon (1937)	Kükenthal (1939)	Govaerts et al. (2018)	
C. brevifolia Cherm.	C. brevifolia Cherm.	C. brevifolia Cherm.	
		C. humbertii Bosser	
C. laxa Cherm.	C. laxa Cherm.	C. laxa Cherm.	
C. laxa var. macrantha Cherm.	C. laxa var. macrantha Cherm.		
	C. elongata (Kunth) Kük.	C. melicoides (Poir.) C.B.Clarke	
C. recurva C.B.Clarke	C. leucocarpa (Ridl.) H.Pfeiff.	C. leucocarpa (Ridl.) H.Pfeiff.	
C. melleri (Baker) C.B.Clarke ex Cherm.	C. melleri (Baker) C.B.Clarke ex Cherm.	C. melleri (Baker) C.B.Clarke ex Cherm.	
C. baronii var. microcarpa Cherm.	C. microcarpa (Cherm.) Kük.	C. microcarpa (Cherm.) Kük.	
	C. natalensis C.B.Clarke	C. natalensis C.B.Clarke	
C. pantopoda (Baker) C.B.Clarke ex Cherm.	C. pantopoda (Baker) C.B.Clarke ex Cherm.	C. pantopoda (Baker) C.B.Clarke ex Cherm.	
	C. pantopoda var. gracilescens Kük.	C. pantopoda var. gracilescens Kük.	
C. baronii C.B.Clarke	C. pantopoda var. baroni (C.B.Clarke) Kük.	C. pantopoda var. baroni (C.B.Clarke) Kük.	
C. baronii var. robusta Cherm.	C. pantopoda var. robusta (Cherm.) Kük.	C. pantopoda var. robusta (Cherm.) Kük.	
C. purpurea Cherm.	C. purpurea Cherm.	C. purpurea Cherm.	
		C. xipholepis (Baker) Henriette & Senterre	
Note:

Seven species of Costularia s.s. were treated by Chermezon (1937), nine species were treated by Kükenthal (1939), and 11 species are currently recognised (Govaerts et al., 2018).

Materials and Methods

Ethics statement

Part of the specimens studied were collected as a part of field expeditions before the 2010 AETFAT conference held in Antananarivo, Madagascar funded by a grant from the Research Foundation–Flanders (FWO) (K204910N), and with support of the Department of Biology, Ghent University, Belgium. Permits to collect and export these specimens were issued by the Madagascar authorities: a collecting permit for Cyperaceae in Madagascar (N°082/10/MEF/SG/DGF/DCB.SAP/SLRSE–Isabel Larridon) was provided by ANGAP Madagascar National Parks authority. The other specimens studied are available in publicly accessible herbaria (BR, G, GENT, K, L, MAU, P, REU, TAN and UPOS; Thiers, 2018).

Nomenclature and taxonomy

A nomenclatural study including the taxonomic history of the genus and its species, critical for the correct coining of the new names and the proper use of prior ones, was performed. The electronic version of this article in portable document format will represent a published work according to the International Code of Nomenclature for algae, fungi and plants (ICN), and hence the new names contained in the electronic version are effectively published under that Code from the electronic edition alone. In addition, new names contained in this work which have been issued with identifiers by IPNI will eventually be made available to the Global Names Index. The IPNI LSIDs can be resolved and the associated information viewed through any standard web browser by appending the LSID contained in this publication to the prefix ‘http://ipni.org/’. The online version of this work is archived and available from the following digital repositories: PeerJ, PubMed Central and CLOCKSS.

Molecular study

All known Costularia s.s. species, except Costularia microcarpa (Cherm.) Kük. which is only known from its type and Costularia brevifolia Cherm. which is rare in collections, were sampled (representing c. 80% of the diversity of the genus, that is, nine out of 11 species and two out of three heterotypic varieties recognised by Govaerts et al. (2018) using multiple accessions per taxon where possible. Additionally, samples were included from several taxa potentially representing new species. The outgroup taxa, selected based on Larridon et al. (2018a), consist of nine species representing the other four genera of the Oreobolus clade of tribe Schoeneae. A total of 36 samples (15 newly sequenced) from 24 different taxa were used for this study. The samples with species names, voucher information, origin and GenBank accession numbers for the sequences, are given in Table S1. The DNA extraction protocol, markers (ETS, ITS and trnL-F), and material and methods for PCR amplification and sequencing and for obtaining alignments follow Larridon et al. (2018a). Sequences were assembled and edited in Geneious R8 (http://www.geneious.com, Kearse et al., 2012), aligned using MAFFT 7 (Katoh, Asimenos & Toh, 2009; Katoh & Standley, 2013) with ‘maxiterate’ and ‘tree rebuilding number’ set to 100 (long run), afterwards, alignments were checked manually in PhyDE 0.9971 (Müller et al., 2010). The alignments used to produce the phylogenies are available as a Data S1.

We first inferred the gene trees for each of the three regions separately to identify potential incongruence. As there were no instances of conflict at well-supported nodes (Figs. S1–S6), the matrices of the three regions were concatenated for the downstream analyses. PartitionFinder 2.1.1 (Lanfear et al., 2012) was used to determine an appropriate data-partitioning scheme from potential partitions that were defined a priori (in this case, each marker was treated as a separate partition), as well as the best-fitting model of molecular evolution for each partition, using the Bayesian Information Criterion. PartitionFinder confirmed the a priori data-partitioning scheme, and the GTR+I+Γ (invgamma) model of sequence evolution was determined to be the best-fitting model for the two nrDNA markers, while the GTR+Γ (gamma) model of sequence evolution was determined to be the best-fitting model for the trnL-F partition in the concatenated data set.

Maximum likelihood (ML) analyses of the optimally partitioned data were performed using RAxML 8.2.10 (Stamatakis, 2014). The search for an optimal ML tree was combined with a rapid bootstrap analysis of 1,000 replicates. Additionally, partitioned analyses were conducted using Bayesian Inference (BI) in MrBayes 3.2.6 (Ronquist et al., 2012). Rate heterogeneity, base frequencies, and substitution rates across partitions were unlinked. The analysis was allowed to run for 100 million generations across four independent runs with four chains each, sampling every 10,000 generations. Convergence, associated likelihood values, effective sample size values and burn-in values of the different runs were verified with Tracer 1.5 (Rambaut & Drummond, 2007). The first 25% of the trees from all runs were excluded as burn-in before making a majority-rule consensus of the 30,000 posterior distribution trees using the ‘sumt’ function. All phylogenetic analyses were run using the CIPRES portal (http://www.phylo.org/; Miller, Pfeiffer & Schwartz, 2011), and were executed for both full and reduced sampling alignments. Trees were drawn using TreeGraph2 (Stöver & Müller, 2010).

Morphological study

Herbarium specimens of BR, G, GENT, K, L, MAU, P, REU, TAN and UPOS (Thiers, 2018) were studied morphologically using a Leica (Leica Microsystems, Wetzlar, Germany) binocular microscope. Measurements where made with a ruler (e.g. leaf and culm length), or using a binocular microscope with graticule (e.g. spikelet and glume length). When measuring width, this was done near the middle of the organ (e.g. middle of the culm). The term peducles represents the main inflorescence branches measured from base of primary inflorescence bract to second order bract. Where possible, links to imaged type specimens are provided (Catalogue des herbiers de Genève, 2018; HerbCat, 2018; Muséum national d’Histoire naturelle, 2018).

Species distributions and conservation assessments

Information on locality data was obtained from the studied herbarium records (see Taxonomic Treatment and Data S2). Georeferenced localities were used to map the distribution of the Costularia species studied in SimpleMappr (Shorthouse, 2010). The extent of occurrence (EOO) and area or occupancy (AOO) of the species were calculated in GeoCAT (Bachman et al., 2011), where the AOO was based on a user defined cell width of two km in line with IUCN Red List criteria (IUCN, 2012). Conservation assessments were prepared according to the guidelines to the IUCN Red List categories and criteria (IUCN, 2012; IUCN Standards and Petitions Subcommittee, 2014).

Results

Molecular study

The multiple-locus BI topology (Fig. 1) did not differ from the multiple-locus ML tree (Fig. S7), except for the sister relationship of clade B. Clade B is sister to clade A in multiple-locus BI topology (Fig. 1), but sister to clade C in the multi-locus ML-analysis (Fig. S7). This relationship is not supported in either result. Four subclades are well supported in the phylogenetic hypothesis (Fig. 1) of the Oreobolus clade of tribe Schoeneae: Costularia (BI posterior probability value 1, ML bootstrap value 100), Chamaedendron (1, 100), Capeobolus + Cyathocoma Nees (1, 100), and Oreobolus R.Br. (0.81). In Costularia, four main clades are well supported: clade A (1, 100), and clade B (1, 100), clade C (1, 100) and clade D (1, 98). In clade A, two species Costularia leucocarpa (Ridl.) H.Pfeiff. + Costularia andringitrensis (formally described in the Taxonomic Treatment) form a supported clade (88) in which Costularia leucocarpa is well supported as a monophyletic lineage (1, 100). These two species are sister to a monophyletic Costularia natalensis (1, 99). In turn, Costularia itremoensis (formally described in the Taxonomic Treatment) is sister to these three species. Clade B consists of the Costularia pantopoda (Baker) C.B.Clarke ex Cherm. species complex with each of the taxa: Costularia baronii C.B.Clarke (1, 95) and Costularia robusta (0.99, 80) (formally recognised at species level in the Taxonomic Treatment) forming well supported monophyletic lineages separate from the typical Costularia pantopoda (Fig. 1). In all analyses, the Costularia baronii and Costularia robusta appear more closely related to each other than to Costularia pantopoda s.s. (Fig. 1; Figs. S1–S7). A last taxon part of this clade, sister to the rest, is a taxon currently identified as Costularia cf. pantopoda. Clade C contains two well supported subclades, one of which (1, 98) includes specimens identified as Costularia laxa Cherm. and as Costularia purpurea Cherm. The latter taxa are supported as monophyletic in some but not all analyses. The second well supported subclade (1, 100) consists of individuals of Costularia melicoides (Poir.) C.B.Clarke. Clade D includes four species: a well supported Costularia xipholepis (Baker) Henriette & Senterre (1, 100), a single accession of Costularia melleri (Baker) C.B.Clarke ex Cherm., and a well supported subclade (1, 90) including Costularia cadetii (formally described in the Taxonomic Treatment) and Costularia humbertii Bosser.

Figure 1 50% majority consensus multiple-locus BI tree with the associated PP values and the bootstrap values of the multiple-locus ML tree.

Only bootstrap values above 70% and posterior probabilities above 0.7 are shown.

Morphological study, species distributions and conservation assessments

Morphological results, species distributions and conservation assessments are elaborated in the Taxomic Treatment. The additional herbarium specimens studied per taxon are listed in Data S2.

Discussion

Four clades are here retrieved in the Oreobolus clade of tribe Schoeneae (Fig. 1): Costularia, Chamaedendron, Capeobolus + Cyathocoma and Oreobolus, in line with recent studies (Larridon et al., 2018a; Semmouri et al., 2018). Viljoen et al. (2013) reconstructed the ancestral areas for tribe Schoeneae but did not obtain a clear result for the ancestral area of the Oreobolus clade. Both Capeobolus and Cyathocoma are found in the Cape Floristic Region, while Chamaedendron is endemic to New Caledonia, and Oreobolus has a wider distribution in the souther hemisphere (Malesia to Australasia, Hawaiian Islands, Costa Rica to Falkland Islands; Govaerts et al., 2018).

Of the four main clades in Costularia, only clade B is restricted to Madagascar, while the others include Madagascar endemics and species found on the Indian Ocean islands and/or mainland Africa (Fig. 1). In clade A, the Madagascan endemic species Costularia leucocarpa and Costularia andringitrensis are sister to Costularia natalensis from southeastern Africa. Costularia itremoensis from South Central Madagascar is sister to these three species (Fig. 1). Clade B consists of the Costularia pantopoda species complex with Costularia baronii and Costularia robusta forming well supported monophyletic lineages separate from the typical Costularia pantopoda (Fig. 1). Costularia robusta was first described as Costularia baronii var. robusta Cherm. This concurs with our results in which Costularia baronii and Costularia robusta are sister species (Fig. 1). A last taxon part of this clade, sister to the others, is currently identified as Costularia cf. pantopoda. This taxon needs further study since it is only known from a single collection with little metadata information. Its morphology appears intermediate between Costularia pantopoda and Costularia itremoensis. Potentially related to clade B is Costularia microcarpa, a species first described by Chermezon (1937) under Costularia baronii (as Costularia baronii C.B.Clarke var. microcarpa Cherm.), and later raised to species level by Kükenthal (1939). Clade C contains two well supported subclades, one of which includes specimens identified as Costularia laxa and Costularia purpurea. These taxa were not always recovered as monophyletic (Figs. S1–S6). Morphological study confirmed that the delimitation between these taxa is unclear, resulting in the decision to combine the two species under a single species name: Costularia purpurea (see Taxonomic Treatment). The second well supported subclade of Clade C consists of individuals of Costularia melicoides (Fig. 1). Costularia melicoides is endemic to the Mascarenes where it is found on both the islands of La Réunion and Mauritius. Clade D includes four species: Costularia xipholepis, a recently rediscovered species endemic to the Seychelles (Henriette et al., 2015), a single accession of Costularia melleri from Central Madagascar, and a subclade including Costularia cadetii and Costularia humbertii. The newly discovered Costularia cadetii and the species Costularia humbertii stand out due to their smaller stature and shorter leaves. Both are restricted to high-elevation zones, but what is remarkable is that while Costularia humbertii is found in the northeast of Madagascar (Marojejy National Park), Costularia cadetii is endemic to La Réunion where it is limited to peaks of the island’s volcanoes. This sister relationship points at a long-distance dispersal event likely from the mountain tops of northeastern Madagascar to those of La Réunion. A species potentially related to Costularia humbertii is Costularia brevifolia with which it shares characters such as a robust caudex, short stature, and short broad leaves, although it is biogeographically (southeastern Madagascar) and ecologically (low-mid elevation) isolated from it (Fig. 2).

Figure 2 Distribution map of Costularia andringitrensis, C. brevifolia, C. humbertii and C. melleri in Madagascar.

The distribution of the species was mapped using SimpleMappr.

Of the 15 species of Costularia recognised here, three-quarters are threatened with extinction because of their restricted distribution ranges and human impact (see Taxonomic Treatment). In Madagascar, habitat destruction and deterioration are the major threats. Additional threats may relate to climate change as some species exclusively occur at (very) high elevation (e.g. Costularia cadetii, Costularia humbertii, Costularia robusta), or to invasive species (e.g. in the Mascarenes). Two species were assessed as critically endangered (CR), six as endangered (EN) and three as Vulnerable (VU) according to IUCN Red List categories and criteria (IUCN, 2012; IUCN Standards and Petitions Subcommittee, 2014). Two endemic but widely distributed species from Madagascar (Costularia leucocarpa, Costularia purpurea) were assessed as least concern (LC), as was Costularia natalensis, the only species occurring in mainland Africa. A final species (Costularia microcarpa) could not be assessed at this time due to lack of information and is considered data deficient (DD). Further research and fieldwork are needed to study the species of Costularia, their populations and the threats they face.

Taxonomic Treatment

Costularia C.B.Clarke in W.H.Harvey & auct. suc. (eds.), Fl. Cap. 7: 274. 1898.

Type: Costularia natalensis C.B.Clarke (lectotype designated by Goetghebeur (1986)).

Perennial herbs, small to tall, tufted or more rarely shortly rhizomatous, caudex sometimes present. Culms scapose or with few nodes. Leaves usually both basal and caudal; basal leaves with poorly defined sheaths; cauline leaves enveloping up to ½ internode length; margins scabrid, spirodistichous, eligulate, blade sometimes deciduous. Inflorescence terminal, (contracted) paniculate with few to numerous spikelets; primary bracts ± leaf-like, sheathing. Spikelets with several distichous, deciduous glumes, of increasing length, the upper (1–)2 glumes each subtending a flower, enclosed by the wings of the next glume. Flowers, lower one (functionally) male (rarely bisexual or absent), upper one bisexual or functionally female (rarely functionally male). Perianth bristles 6, fimbriate to ciliate, mostly longer than the nutlet and deciduous with it. Stamens 3. Style trifid, style base often distinct (at anthesis), thickened, persistent, often scabrid. Nutlet ovoid or oblong, rounded trigonous, often 3-ribbed, ± stipitate, beaked, surface smooth or rugulose.

Includes: 15 species.

Distribution: southeastern Africa (Malawi, Mozambique, South Africa, Swaziland, Zimbabwe), Madagascar, the Mascarenes (La Réunion, Mauritius), and the Seychelles (Mahé).

Key to the species of Costularia

1 Plants <30 cm tall with flowering culm scarcely exceeding the basal leaves; cauline leaves absent2

1 Plants >30 cm tall with flowering culm exceeding the leaves; cauline leaves present3

2 Spikelets with lower flower male, upper flower bisexual (endemic to Andringitra Mountains, Madagascar)1. Costularia andringitrensis

2 Spikelets with two bisexual flowers (endemic to La Réunion)4. Costularia cadetii

3 Basal leaves conspicuously shorter than the flowering culm with leaf blades 7–15 cm long, apex rounded-obtuse4

3 Basal leaves not conspicuously short compared to the flowering culm with leaf blades >15 cm, apex generally long tapering5

4 Caudex four to five cm wide; basal leaves spirodistichous; leaf blades 7–12 mm wide; pedicels of the spikelets 5–20 mm long (endemic to SE Madagascar)3. Costularia brevifolia

4 Caudex one to two cm wide; basal leaves distichously and flabellately inserted on the caudex; leaf blades five to eight mm wide, sickle-shaped; pedicels of the spikelets one to four mm long. (endemic to NE Madagascar)5. Costularia humbertii

5 Spikelets up to 3.8–5 mm long7. Costularia leucocarpa

5 Spikelets >5.5 mm long6

6 Flowers 2, lower bisexual, upper male (or rarely only one flower) (Madagascar, Mascarenes)7

6 Flowers 2, lower male, upper bisexual (Africa, Madagascar, Seychelles)8

7 Pedicels of the spikelet erect; glumes reddish-black, with colourless-whitish margins; nutlet smooth (La Réunion, Mauritius)8. Costularia melicoides

7 Pedicels of the spikelet generally curved; glumes entirely (dark) purple; nutlet rugulose- reticulate (Madagascar)13. Costularia purpurea

8 Glumes 16–18 per spikelet9. Costularia melleri

8 Glumes 5–14 per spikelet9

9 Culm <1 mm wide; leaf blades ≤1.5 mm wide12b. Costularia pantopoda var. gracilenscens

9 Culm ≥ 1.5 mm wide; leaf blades wider than 1.5 mm10

10 Culms c. six mm wide11

10 Culms 1.5–5 mm wide12

11 Caudex not present; glumes 8–12 per spikelet, straw-coloured to purplish striate (endemic to SE Madagascar)10. Costularia microcarpa

11 Caudex strongly developed and long; glumes 12–14 per spikelet, purplish black (endemic to N Madagascar)14. Costularia robusta

12 Vegetative culm 17–70 cm × 1.7–3.5 mm; cauline leaves 1–213

12 Vegetative culm 50–150 cm × 2.5–5 mm; cauline leaves 2–514

13 Peduncles longest 5.5–11 cm; empty glumes 6–116. Costularia itremoensis

13 Peduncles longest four to five cm; empty glumes 3–612a. Costularia pantopoda var. pantopoda

14 Leaf blades 70–125 cm × 7–10 mm (Seychelles)15. Costularia xipholepis

14 Leaf blades 30–80 cm long × 2–8 mm (Madagascar)15

15 Pedicels of the spikelets ≥5 mm long; glumes largest 4–5.5 mm long13. Costularia purpurea

15 Pedicels of the spikelets one to six mm long; glumes largest 5.5–7 mm long16

16 Peduncles long (longest up to c. 15 cm); spikelets oblong; glumes largest 6–7 mm long (SE Africa)11. Costularia natalensis

16 Peduncles short (longest c. five to eight cm); spikelets lanceolate; glumes largest 5.5–6 mm long (endemic to Madagascar)2. Costularia baronii

1. Costularia andringitrensis Larridon sp. nov.—Figs. 2–4

Type. Madagascar, Fianarantsoa, Haute Matsiatra, Andringitra National Park, Diavolana Trail, 22°07′28.0″S, 46°52′32.7″E, 2,063 m, 18 April 2010, I. Larridon, W. Huygh, M. Reynders, A.M. Muasya & V. Randrianasolo 2010-0140 (holotype TAN!, isotypes BOL!, GENT!).

Diagnosis: Costularia andringitrensis differs from all other Costularia species from Madagascar by its small stature with the flowering culm scarcely exceeding the leaves. In this aspect it mostly resembles Costularia cadetii from La Réunion from which it can easily be distinguished by the latter maturing two nutlets per spikelet.

Small perennial herb, flowering culm up to 24 cm, scarcely exceeding the leaves. Caudex absent. Culm (excluding the inflorescence) short and slender, 5–7.2 cm × 1.1–1.2 mm. Basal leaves distichous, bases of old burnt leaves can be present; leaf sheaths 1.5–2 cm × up to 4 mm, only slightly wider than the leaf blade, indistinct, straw-coloured to green; leaf blades linear, flat, 8–34 cm × 1.2–2.6 mm, margins scabrid. Cauline leaves absent. Inflorescence a contracted panicle, 12–19 × 0.5 cm, composed of few to several spikelets; inflorescence bracts 6, unequal, sheathing, dark reddish brown, margins scabrid; longest bract 12.5–15 cm × 2.5 mm. Peduncles unequal, up to 2.6 cm long, margins smooth to scabrid. Pedicels of the spikelets unequal, one to five mm long, minutely papilose, margins scabrid. Spikelets lanceolate, (4–)5–5.5 × 1.1–2 mm, dark purple. Glumes distichous, narrowly ovate, boatshaped, acuminate (upper glumes) to long mucronate (up to c. one mm, lower glumes), 3–4 × 1.5–2 mm, dark purple on upper part including mucro if present and pale brown on lower part, margins scabrid; three lower glumes empty, two upper glumes fertile. Flowers 2, lower male, upper bisexual. Perianth bristles 6, pale, thin, antrorsely ciliate, up to 13 mm long. Stamens 3. Style deeply trifid. Immature nutlet (see Fig. 3D) rounded trigonous with distinct bulbous style base remaining; ripe nutlets not studied as they were already shed from plants in all available specimens.

Figure 3 Illustration of Costularia andringitrensis (Larridon et al. 2010-0140 GENT).

(A) Habit; (B) inflorescence; (C) spikelet; (D) bisexual flower and glume; (E) male flower and glume. Scale bars: A–B = three cm; C–E = three mm. Illustration drawn by Juliet Beentje.

Figure 4 Habitat and morphology of Costularia andringitrensis.

(A) Plant in situ; (B) imaged plant. Photos taken by Wim Huygh in Adringitra National Park, Madagascar on 18 April 2010.

Distribution

The species is only known from south-central Madagascar, where it was found in the Andringitra National Park, Haute Matsiatra region, Fianarantsoa province (Fig. 2).

Ecology

This species is found in near rocks in grassland to ericoid shrubland vegetation at 2,000–2,500 m in elevation.

Phenology

Immature inflorescence observed in November, while the specimen collected in April had already shed its ripe nutlets.

Etymology

The species is named for the Andringitra National Park in Madagascar.

Conservation status

Costularia andringitrensis is a small perennial herb endemic to Madagascar, where it is only known from two specimens and occurs in a restricted area in the Andringitra National Park. It is only known from a single location and a minimum AOO of eight km2. However, there are other potential areas of occurrence for the species that have not yet been explored. The species is threatened by cattle grazing and by fires started for pastoral reasons which can easily get out of control and enter the National Park (I. Larridon, 2010, personal observation; F. Rakotonasolo, 2017, personal observation). Therefore, it is assessed as Critically Endangered: CR B2ab(ii,iii).

Notes

As is commonly seen in tropical Cyperaceae species occurring at high elevation (I. Larridon, 2010, personal observation), Costularia andringitrensis is characterised by very dark spikelets. In the molecular phylogenetic hypothesis (Fig. 1), it is retrieved as sister to Costularia leucocarpa.

Although species of tribe Schoeneae are adapted to natural fire, if fire frequency is increased, especially by herders, this can threaten their regeneration (A.M. Muasya, 2010, personal observation). However, complete absence of fire can also be a threat as most species occur in habitats where open/forest are alternative states. Forests are kept out by the fire, whose absence could lead to forest encroachment. Most species of tribe Schoeneae are shade intolerant and thus would die if shaded.

2. Costularia baronii C.B.Clarke in W.H.Harvey & auct. suc. (eds.), Fl. Cap. 7: 274 (1898) ≡ Costularia pantopoda var. baronii (C.B.Clarke) Kük., Repert. Spec. Nov. Regni Veg. 41: 67 (1939)—Figs. 5, 6

Figure 5 Distribution map of Costularia itremoensis, C. microcarpa and C. pantopoda in Madagascar.

The distribution of the species was mapped using SimpleMappr.

Figure 6 Habitat and morphology of Costularia baronii.

(A) Plant in situ; (B) inflorescence detail. Photos taken by Wim Huygh in Adringitra National Park, Madagascar on 18 April 2010.

Type (lectotype designated here). Madagascar, Central Madagascar, R. Baron 3316 (lectotype: K000244885!, isolectotype: MNHN-P-P00459989!).

Robust perennial herb: Culms 20–80 cm × c. 5 mm, generally quite robust. Basal leaves with leaf blades 40–80 cm × 2–6 mm wide, flat. Cauline leaves 2. Inflorescence a narrow, tight, very upright panicle with numerous spikelets; inflorescence bracts 8–11. Peduncles erect, the longest five to eight cm. Pedicels of the spikelets erect, one to five mm long. Spikelets lanceolate, 6–8 × 1.5–2 mm. Glumes 8–12, reddish brown to black, lanceolate, subobtuse, strongly distichous, the largest 5.5–6 mm long, lower 6–10 empty; empty glumes much smaller than the fertile glumes. Nutlet subglobose, weakly trigonous, 2.25 mm long, rugolose, greyish green; beak 0.75 mm long, obtuse, not depressed at the base, almost as wide as the nutlet.

Distribution

Costularia baronii occurs in the Antananarivo, Fianarantsoa and Toliara provinces of Madagascar (Fig. 5).

Ecology

It has been found growing in rocky areas (e.g. rock crevices along a stream bank), ericoid shrubland at elevations of 1,300 to almost 2,200 m.

Phenology

Flowering specimens were collected from March to May, fruiting plants in October, while plants collected in December and January had either shed their nutlets or bore very young inflorescences.

Conservation status

Costularia baronii is distributed in the Antananarivo, Fianarantsoa and Toliara provinces of Madagascar, and occurs in at least four protected areas, that is, Andringitra, Ankaratra Massif, Andohahela, Ibity Massif and Pic d’Ivohibe. Threats to this taxon need further investigation but in the Andrigitra National Park, its habitat and area of occupancy are impacted negatively by cattle grazing and by fires started for pastoral reasons which can easily get out of control and enter the National Park (I. Larridon, 2010, personal observation; F. Rakotonasolo, 2017, personal observation). Based on 10 georeferenced herbarium specimens, the species occurs in at least seven locations and has an estimated AOO of 36 km2 and an EOO of 16,292 km2. Using IUCN criteria, it can be assessed as VU B1ab(ii,iii)+2ab(ii,iii).

Notes

Chermezon (1937) identified Humbert 7008 as Costularia pantopoda var. pantopoda. However, we believe this specimen better fits with the description of Costularia baronii.

3. Costularia brevifolia Cherm., Bull. Soc. Bot. France 69: 723. 1922 publ. 1923. ≡ Tetraria brevifolia (Cherm.) T.Koyama, J. Fac. Sci. Univ. Tokyo, Sect. 3, Bot. 8: 74. 1961—Figs. 2, 7

Figure 7 Illustration of Costularia brevifolia (Razakamalala 4866 K).

(A) Habit; (B) leaf tip detail; (C) lower leaf detail; (D) detail of inflorescence; (E) male flower and glume. Scale bars: A = three cm, B–D = one cm, E = five mm. Illustration drawn by Juliet Beentje.

Type (lectotype designated here). Madagascar, Toliara, Mananara Bassin, 700 m, June 1919, H. Perrier de la Bâthie 12643 (lectotype: MNHN-P-P00459974!; isolectotypes: MNHN-P-P00459972!, MNHN-P-P00459973!).

Robust perennial herb: Caudex 10–12 cm × 4–5 cm. Culm (appearing) lateral, robust, 50–80 cm × 5–8 mm, smooth, with obtuse edges, slightly compressed. Basal leaves spirodistichously inserted on the caudex, leaf sheaths, 3–4 cm × 3–4 cm, brown, shiny, margins scarious, at the apex abruptly contracted, old sheaths fibrous, leaf blades 7–10 cm × 7–12 mm, flat, leathery, margins scabrid, revolute, apex rounded-obtuse. Cauline leaves 1–3, far apart, sheathing, sheaths brown. Inflorescence a panicle c. 45 cm long, loosely compound. Peduncles unequal, up to 7.5 cm long. Pedicels of the spikelets suberect or curved, 5–20 mm long. Spikelets oblong-lanceolate, compressed, apex subacute, 7–8 mm × 1.5–2 mm. Glumes distichous, oblong-lanceolate, five to six mm long, densely imbricate, straw-coloured to brown, purple-tinged, edges only from the keel up sparsely ciliolate, prominently acute or mucronate, lower three to four glumes empty, two upper glumes fertile. Flowers 2, lower male, upper bisexual. Perianth bristles 6, pale brown, plumose, three times longer than the nutlet. Stamens 3, filaments reddish-brown, anthers linear, connective conical-subulate, purple. Style long, deeply trifid, pale, with a triangular thickly cone-shaped persistant base. Nutlet 1.5 mm long, brown, obovate-oblong, with an attenuate base.

Distribution

Endemic to southeastern Madagascar and only known from the Atsimo Atsinana and Anosy regions in the Fianarantsoa and Toliara provinces (Fig. 2).

Ecology

It has been found growing on humid rocks in peatlands, on laterite and granite in tropical forest, and in faults of gneiss rock escarpments, at elevations of (200–)600–900 m.

Phenology

Flowering specimens were collected from in March. Young inflorescences can be observed on the specimens collected in February, while old inflorescences remain on the plants until October–November.

Conservation status

Costularia brevifolia is a robust perennial herb, limited in distribution to the forested mountain ranges of south-eastern Madagascar at mid-elevation. It is known from only four locations. The estimated EOO is 2,463 km2 and the area of occupancy is 20 km2. According to the limited metadata available this species likely occurs in the protected areas of Midongy du Sud and Andohahela. Fire (natural or man-made) and disturbance or elimination as a result of deforestation for agricultural extension are the major threats wich affect this species. Hence, it is assessed as EN B1ab(i,ii,iii,iv)+B2ab(i,ii,iii,iv).

Notes

One of only two short-leaved Costularia species in Madagascar; the other being Costularia humbertii. Costularia brevifolia is endemic to southeastern Madagascar, while Costularia humberti is endemic to the Marojejy National Park in northeastern Madagascar. Although both are likely related based on morphological resemblance, amplification of DNA extracted from the limited material available of Costularia brevifolia was unsuccessful, so a close relationship between the two short leaved species remains unconfirmed.

4. Costularia cadetii Larridon sp. nov. —Figs. 8–10

Figure 8 Illustration of Costularia cadetii.

(A) Habit (Luceño & Guzmán 4ML09 UPOS); (B) inflorescence (Luceño & Guzmán 4ML09 UPOS); (C) detail of inflorescence (Márquez-Corro et al. 04JMC17 UPOS); (D) spikelet (Márquez-Corro et al. 04JMC17 UPOS); (E) bisexual flower (Márquez-Corro et al. 04JMC17 UPOS); (F) anther (Luceño & Guzmán 4ML09 UPOS); (G) nutlet (Márquez-Corro et al. 04JMC17 UPOS). Scale bars: A–B = three cm, C–E = three mm, F = one mm, G = three mm. Illustration drawn by Juliet Beentje.

Figure 9 Habitat and morphology of Costularia cadetii.

(A) Habitat; (B) habit; (C) inflorescence; (D) base of plant. Photos taken by Jeremy Bruhl (A, C, D) in La Réunion at Nez Coupé de Sainte-Rose on 4 January 2011 and by Modesto Luceño (B) in La Réunion at Piton de la Fournaise-Pas de Bellocombe on 1 January 2009.

Figure 10 Distribution map of Costularia cadetii and C. melicoides in La Réunion and Mauritius (Mascarene Islands).

The distribution of the species was mapped using SimpleMappr.

Type. LA RÉUNION, Saint-Benoît, Sainte-Rose, Pas de Bellecombe, 21°13′21.38″S, 55°41′17.27″E, 2,328 m, 6 March 2017, J.I. Marquez-Corro et al. 04JMC17 (holotype K!, isotypes UPOS!).

Diagnosis: This species is closely related to Costularia humbertii from northern Madagascar, from which it differs in its smaller habit, absence of a caudex, the basal leaves equaling or overtopping the flowering culm, and having two bisexual flowers. It can be distinguished from the only other species of Costularia on La Réunion by its much smaller habit and having two bisexual flowers.

Small perennial herb, flowering culm up to 28 cm, scarcely exceeding the leaves. Caudex absent or short (c. 0.5 mm wide). Culm slender, 4–12.5 cm × 1.7–1.9 mm. Basal leaves distichous; leaf sheaths 2.5–2.8 cm × 6–7 mm, reddish-purplish brown; leaf blades linear, flat, 8.5–29 cm × 1.4–4 mm, scabrid on the margins. Cauline leaves absent. Inflorescence a somewhat contracted panicle, 9–15.5 × 1 cm, composed of numerous spikelets; inflorescence bracts 5, unequal, sheathing, dark reddish brown, scabrid on the margins; longest bract 8.5–13 cm × 2–3 mm. Peduncles unequal, up to four cm long, margins scabrid at least near the apex. Pedicels of the spikelets unequal, 1–12 mm long, minutely papilose, margins scabrid. Spikelets lanceolate, 4.5–5 × 1.1–2 mm, dark purple. Glumes distichous, narrowly ovate, boatshaped, acuminate to long mucronate (up to c. one mm), 3.5–4.5 × 1.8 mm, dark purple with pale lower third and pale mucro, scabrid to minutely ciliate on the margins, keel and top half of abaxial surface; two to three lower glumes empty, two upper glumes fertile. Flowers 2, both bisexual. Perianth bristles 6, pale, antoresly ciliate. Stamens 3, anthers linear, one to two mm long with short conical connective. Style deeply trifid. Nutlet rounded trigonous, obovate, dark brown with three pale bands on the ridges, base attenuate 1.3–1.5 × 0.8–0.9 mm.

Distribution

Costularia cadetii is a small perennial herb, endemic to La Réunion and found only in the Parc National de La Réunion at elevations of 1,700–2,400 m (Fig. 10).

Ecology

Found growing in rocky areas, montane grasslands and ericoid vegetation close to volcanic crater edges at high elevation.

Phenology

Flowering specimens were collected in January, fruiting specimens in February and March. The specimens collected by Cadet in May had shed their glumes and nutlets, while the plants collected in November and December were vegetative or immature.

Etymology

The first record of this species (Cadet 454) was collected by in 1965, and on its label the following note is written ‘Costularia sp. Further material needed!’. Thérésian Cadet (1937–1987) was a botanist from La Réunion specialised in the vegetation from the Mascarene Islands. He taught plant biology at the University of La Réunion and was one of the main authors of the Flore des Mascareignes. This species is named in his honour.

Conservation status

Costularia cadetii is a small perennial herb, endemic to La Réunion. It is known only from three locations within the Parc National de La Réunion. The area, extent and quality of habitat of this species is threatened by fire, volcanic activity and climate change. Based on the seven known herbarium collections, the minimum estimated area of occupancy is 20 km2 and the minimum estimated EOO is 250 km2. It is hence categorised as EN B1ab(iii)+B2ab(iii).

Notes

Although not closely related to it, morphologically, Costularia cadetii most closely resembles the Madagascan endemic species Costularia andringitrensis, from which it can easily be distinguished by the former maturing two nutlets per spikelet.

5. Costularia humbertii Bosser, Naturaliste Malgache 7: 121. 1955—Figs. 2, 11

Figure 11 Illustration of Costularia humbertii (Miller & Lowry II 4175 GENT).

(A) Habit; (B) detail of inflorescence; (C) spikelet; (D) male flower; (E) anther; (F) bisexual flower and glume. Scale bars: A = three cm, B–C = one cm, D–F = three mm. Illustration drawn by Juliet Beentje.

Type (lectotype designated here). Madagascar, Antsiranana, Marojejy, 1,850–2,137 m, 26 March 1949—2 April 1949, H. Humbert & G. Cours 23708 (lectotype: MNHN-P-P00459980!; isolectotypes: MNHN-P-P00459978!, MNHN-P-P00459979!, G00406272!).

Robust perennial herb: Caudex robust, 5–10 cm × 1–2 cm. Culm 30–70 cm × 2–3 mm, compressed, smooth. Basal leaves distichous, flabellately arranged; leaf sheaths densely imbricate, two to three cm long, reddish brown to chestnut coloured; leaf blades leathery, falciform (sickle-shaped), flat, canaliculate, pale green, minutely papillose above, 7–15 cm × 5–8 mm, much shorter than the culm, apex subacute to rounded-obtuse, margins scabrid. Cauline leaves 1–3, sheathing. Inflorescence a somewhat lax and compound panicle, foliate, 15–25 cm long, composed of five to seven erect to flexuous fascicles. Peduncles unequal, at most seven cm long, margins scabrid, papillose above. Pedicels of the spikelets one to four mm long, green. Spikelets lanceolate, 5–7.5 mm long, dark purple. Glumes 4–6, ovate, distichous, 1-veined, 4–5 × 1 mm, margins minutely ciliolate, keel somewhat scabrid, apex acute to mucronate, two to four lower glumes empty, two upper glumes fertile. Flowers 2, lower male, upper bisexual. Perianth bristles 6, longer than the nutlet, shortly ciliate. Stamens 3, anthers long and linear, apiculate, three mm long. Style deeply trifid. Nutlet 2.5 mm long, smooth, castaneous, trigonous, base attenuate, beak pale, one mm, ciliolate.

Distribution

Endemic to the Antsiranana province of Madagascar where it is restricted to the high-elevation zone of the Marojejy National Park (Fig. 2).

Ecology

Found growing in swamps in high elevation ericoid vegetation, and on gneiss and quartzite rocks of the mountain ridge, at elevations of 1,400–2,200 m.

Phenology

Flowering specimens were collected from March to early April. Young inflorescences can be observed on the specimens collected in November–December.

Conservation status

Costularia humbertii is endemic to the Antsiranana province of Madagascar and is limited in distribution to the high-elevation zone of the Marojejy National Park. The minimal area of occupancy was calculated as 24 km2, the estimated EOO is 17 km2 and the species is only known from one location. Fire (natural and man-made) and disturbance of its habitat as a result of logging, firewood collection and charcoal are the major threats which may affect this species. Costularia humbertii is only known from seven herbarium collections and has not been collected since 1989. Research is needed to investigate its current status at the single known location. Here, we assess the species as CR B1ab(i,ii,iii).

Notes

One of two short leaved Costularia species in Madagascar, the other being Costularia brevifolia. In the molecular phylogenetic hypothesis (Fig. 1), Costularia humbertii appears to be closely related with a small high-elevation species from La Réunion (Costularia cadetii).

6. Costularia itremoensis Larridon sp. nov. —Figs. 5, 12

Figure 12 Illustration of Costularia itremoensis.

(A) Habit (Humbert 30060 P); (B) spikelet (Humbert & Swingle 4995 P); (C) bisexual flower and glume (Humbert & Swingle 4995 P); (D) male flower and glume (Humbert & Swingle 4995 P); (E) anther and filament, immature (Humbert & Swingle 4995 P). Scale bars: A = three cm, B–D = five mm, E = three mm. Illustration drawn by Juliet Beentje.

Type. Madagascar, Fianarantsoa, Isalo Plateau, W of Ranohira, sandstone rocks, 800–1,000 m, 30 July 1928, H. Humbert & C.F. Swingle 4995 (holotype: MNHN-P-P0318446!, isotypes: K!, TAN).

Diagnosis: This species resembles most closely Costularia pantopoda var. pantopoda from which it can be distinguished by having longer peduncles (longest 5.5–11 vs. 4–5 cm) and more emptyglumes (6–11 vs. 3–6).

Medium-sized to tall perennial herb, up to c. 1.4 m. Culm 17–68 cm × 1.7–3.5 mm. Basal leaves distichous; leaf sheaths 2–6 cm × 7–11 mm, reddish-brown, sometimes burnt old leaf bases present; 20–70 cm × 2.5–5.5 mm, flat, margins scabrid. Cauline leaves 1–2, margins scabrid, sheaths brownish. Inflorescence a panicle, somewhat contracted when young, but more lax at maturity, 24–68 cm long; inflorescence bracts 6–8, unequal, up to four mm wide, sheating, reddish, margins scabrid. Peduncles longest 5.5–11 cm, unequal, flattened, margins scabrid. Pedicels of the spikelets erect, 2–11 mm, margins scabrid. Spikelets oblanceolate, (4.5–)5.5–10 × 1.2–2.8 mm. Glumes 8–13, distichous, the largest 4–6.5 mm long, dark purple above pale below; lower 6–11 glumes empty, acute, increasing in length; upper 2 glumes fertile, more obtuse. Flowers 2, lower male, upper bisexual. Perianth bristles 6, plumose, long antrorsly ciliate. Stamens 3; anthers 6–6.5 mm, linear. Style trifid, long. Nutlet rounded trigonous, 2.4 × 1.2 mm; beak c. 0.5 mm, pale, ciliate.

Distribution

Costularia itremoensis is endemic to Madagascar and is found in the highlands of South Central Madagascar, in the Fianarantsoa province (Fig. 5).

Ecology

The habitat in which this species is found consists of bare rocks and/or grassland in the Itremo massif (L. Rabarivola, 2014, personal observation). In Isalo, its habitat is dominated by wooded grassland-bushland mosaic and/or plateau grassland-wooded grassland mosaic (Moat & Smith, 2007) between 800 and 1,700 m in elevation.

Phenology

Flowering/fruiting specimens were collected from July to September, plants collected from January to April were immature.

Conservation status

Costularia itremoensis is endemic to Madagascar and is found in the highlands of South Central Madagascar, in the Fianarantsoa province. Based on the limited metadata available it likely occurs in the Itremo new protected area and Isalo National Park. The estimated EOO was calculated as 7,169 km2 and the minimal area of occupancy is 20 km2. This species is only known from three locations and is threatened by grazing and uncontrolled fire from pastures fire. Its habitat is also threatened by deforestation from logging, firewood collection and mining. Therefore, this species is assessed as Endangered: EN B2ab(i,ii,iii).

Notes

Costularia itremoensis is sister to a clade including Costularia leucocarpa + Costularia andringitrensis and Costularia natalensis (Fig. 1).

7. Costularia leucocarpa (Ridl.) H.Pfeiff., Repert. Spec. Nov. Regni Veg. 23: 346. 1927 ≡ Rhynchospora leucocarpa Ridl., J. Linn. Soc., Bot. 20: 335. 1883 ≡ Costularia recurva C.B.Clarke, Ill. Cyper.: t. LXXXVIII (1909), nom. superfl. ≡ T. leucocarpa (Ridl.) T.Koyama, J. Fac. Sci. Univ. Tokyo, Sect. 3, Bot. 8: 75. 1961—Figs. 13, 14

Figure 13 Illustration of Costularia leucocarpa (Larridon et al. 2010-0237 GENT).

(A) Habit; (B) detail of inflorescence; (C) flowering spikelet (some glumes already fallen off); (D) bisexual flower; (E) male flower; (F) nutlet. Scale bars: A = three cm, B = five mm, C–F = three mm. Illustration drawn by Juliet Beentje.

Figure 14 Distribution map of Costularia leucocarpa and C. purpurea in Madagascar.

The distribution of the species was mapped using SimpleMappr.

Type (lectotype designated here). Madagascar, Central Madagascar, R. Baron 399 (lectotype K000244883!; isolectotypes BM, K!, MNHN-P-P00459985!).

= Cladium fimbristyloides Baker, J. Linn. Soc., Bot. 22: 531. 1887. Type (lectotype designated here). Madagascar, Central Madagascar, R. Baron 4193 (lectotype K000244884!; isolectotype MNHN-P-P00459986!).

Perennial herb up to c. one m tall with a woody rhizome (c. four mm diam.), caudex sometimes present (c. seven mm diam.) covered in old leaf sheaths. Culm strong but slender towards the apex, 38–60 cm × 1.9–2.6 mm diam., compressed to obtuse angled, grooved, minutely papillose. Basal leaves many, distichous; leaf sheaths brown-purplish, 3.5–4.5 cm long; leaf blades long acuminate, flat, margins scabrid. Cauline leaves 2, longest up to c. 35 cm × 3 mm, keeled, sheaths long somewhat enlarged, purplish, mouth obliquely cut. Inflorescence a semi-compound panicle, 40–70 cm long, narrow, lax, built up out of eight to nine widely spaced fascicles; inflorescence bracts leafy and much overtopping the fascicles, sheath long and brown-purplish. Peduncles unequal, up to c. 10 cm long, flattened, scabrid. Pedicels of the spikelets (2–)4–10 mm long, arched recurved, scabrid. Spikelets oblong-lanceolate, subterete, 3.8–5 × 2 mm. Glumes (5–)6(–7), distichous, ovate, subobtuse, above dark purplish, below straw-coloured, nerveless except the keel, finely ciliolate, 3–3.5 × 1.7–2.4 mm; three to five lower glumes empty, increasing in size; two upper glumes fertile. Flowers 2, lower male, upper bisexual. Perianth bristles 6, ± as long as the nutlet including its beak, tender, pale brown, antrorsely dense and shortly ciliolate, not plumose. Stamens 3, filaments and anthers reddish, anthers linear, connective short, wide pyramidal, dark purplish. Style rigid, brown, trifid, thickened at base, swollen in the middle, triquetrous, dark-purple, hairy, persistent. Nutlet 2.3–2.7 × 1.5–1.7 mm swollen-trigonous, pale, bright, smooth, hardly furrowed; beak narrow, 2–2.5 mm long.

Distribution

Endemic to Madagascar, found in the provinces Antananarivo, Antsiranana, Fianarantsoa, Toamasina and Toliara (Fig. 14).

Ecology

The species occurs at mid to (very) high elevation, and has been collected along mountain ridges, from thickets on rock formations, and in open forest.

Phenology

Flowering specimens were collected in December–January, while fruiting specimens were collected in February and March.

Conservation status

Costularia leucocarpa is endemic to Madagascar and found in Antsiranana, Antananarivo, Toamasina, Fianarantsoa and Toliara provinces, where it has been collected along mountain ridges, from thickets on rock formations. The species occurs in Ranomafana National Park, Tsaratanana Reserve Naturelle Intégrale and Manongarivo Special Reserve. The species has a large distribution range (AOO = 124 km2) and its estimated EOO is 7,636 km2, which is much larger than the threshold for a threatened category. Despite its habitat being under various anthropogenic pressures, Costularia leucocarpa is here assessed as LC because (1) no specific threats to its survival have been observed, (2) it is widely distributed in Madagascar, and (3) occurs in several protected areas.

Notes

Since Costularia recurva shares syntypes with the older name Costularia leucocarpa, both can be lectotypified to the same specimen (Baron 399 K000244883) rendering Costularia recurva superfluous.

Previously, the number and position of flowers has been unclear. Chermezon (1937) (in general for the genus) and Kükenthal (1939) (for Costularia leucocarpa) described the male flower to be born by the third glume from the top of the spikelet, the bisexual flower to be born by the second glume from the top, and the topmost glume to be empty and reduced. At first glance, this appears correct, but when comparing Costularia leucocarpa spikelets with those of the other Costularia species where the topmost glumes are fertile and the lower glumes are sterile, and taking in consideration the common metatopic displacement (epicaulescence) of the glumes and flowers on the rachilla in spikelets with distichous glumes of species of Cyperaceae subfamily Cyperoideae (Vrijdaghs et al., 2010, 2011), we believe that Costularia leucocarpa represents the common pattern observed in the rest of the genus.

8. Costularia melicoides (Poir.) C.B.Clarke, Bull. Misc. Inform. Kew, Addit. Ser. 8: 48 (1908). ≡ Cyperus melicoides Poir. in J.B.A.M.de Lamarck, Encycl. 7: 273 (1806) ≡ Machaerina melicoides (Poir.) Bojer, Hortus Maurit.: 386 (1837) ≡ Asterochaete elongata Kunth, Enum. Pl. 2: 312 (1837) ≡ S. elongatus Willd. ex Kunth, Enum. Pl. 2: 312 (1837), nom. inval. ≡ Carpha elongata (Kunth) Boeckeler, Linnaea 38: 273 (1874) ≡ Cyclocampe elongata (Kunth) Benth. & Hook.f., Gen. Pl. 3: 1063 (1883) ≡ Lophoschoenus elongatus (Kunth) H.Pfeiff., Beih. Bot. Centralbl. 44(1): 133 (1927) ≡ Costularia elongata (Kunth) Kük., Repert. Spec. Nov. Regni Veg. 44: 187 (1938), nom. illeg. ≡ T. elongata (Kunth) T.Koyama, J. Fac. Sci. Univ. Tokyo, Sect. 3, Bot. 8: 74 (1961)—Fig. 10

Type (lectotype designated here). Mauritius, L.M.A. du Petit Thouars s.n. (herb. Willd. 1115 fol. 1) (lectotype MNHN-P-P00552880!, isolectotype MNHN-P-P02284597!).

= Carpha costularioides C.B.Clarke, Bull. Misc. Inform. Kew, Addit. Ser. 8: 43 (1908) (earlier as Carpha aubertii Nees var. explicatior C.B.Clarke, Consp. Fl. Afr. 655 (1894), nom. inval. with mention of type but no description) ≡ Costularia explicatior Cherm., Bull. Soc. Bot. France 69: 722 (1922). Lectotype (designated here). Mauritius, Flacq, Le Grand Fond, 280 m, 17 June 1890, H.H. Johnston s.n. (lectotype K000244879!, isolectotype MAU0003574!).

Perennial herb with short rhizome with stiff fibres. Culm 35–100 cm × 1.5–4 mm, striate, minutely puncticulate. Basal leaves crowded, distichous; leaf sheaths four to six cm long, indistinct, straw-coloured to purple, multiveined; leaf blades c. 27–60 cm × 2.5–5 mm, flat, indistinctely keeled, tapered at the tip, edges minutely serrulate. Cauline leaves 3–4, very distant; leaf sheaths, long, green-purplish, mouth oblique. Inflorescence an elongate panicle, 30–85 cm long, with c. 9–11 partial inflorescences, distantly spaced; inflorescence bracts longer than the partial inflorescence they subtend, sheaths purplish. Peduncles unequal, up to c. 12 cm. Pedicels of the spikelets erect, flattened, margins slightly scabrid, 4–15 mm long. Spikelets oblong-lanceolate, 5.5–7.5 × 1.5–2 mm, somewhat flattened. Glumes 5–8, distichous, lanceolate-ovate, acuminate, reddish-black, with colourless-whitish margins; lower glumes empty, scabrid on the midvein, mucronatae-aristulate; two upper glumes fertile, barely mucronate; rhachilla short and erect. Flowers 2, lower bisexual, upper male. Perianth bristles 6, longer than the nutlet, pale to rusty-coloured, antrorsely densely ciliate-scabrid. Stamens 3; anthers linear yellow; connective short, bent, purple. Style trifid, base elongate-conical, triquetrous, pale, margins hispidulous, persistent. Nutlet swollen-trigonous, longitudinaly trisulcate, pale, smooth, 2–2.3 mm long, base long cuneate; beak narrow, 2–2.5 mm long.

Distribution

Endemic to the Mascarene Islands of La Réunion and Mauritius (Fig. 10).

Ecology

Costularia melicoides prefers mid to higher elevation on the island of La Réunion: (500–)900–1,700 (–2,000) m where it occurs in ericoid thickets (avounes), moist tropical forest, forest with Acacia heterophylla (tamarinaie), and humid tickets with Pandanus (C. Fontaine, 2018, personal communication). However, in Mauritius, it is found on boulders or in clumps in seasonally-flooded upland marshes near Petrin in the Black River Gorges National Park at elevations of c. 600–700 m, in upland marshes and thickets in Perrier Nature Reserve at c. 550 m in elevation, and in the district Flacq it was found at an elevation of 280 m.

Phenology

Flowering specimens were collected in February (La Réunion) and June (Mauritius), fruiting specimens were collected in April and May and from October to January (La Réunion).

Conservation status

Costularia melicoides occurs in the four regions of La Réunion, and it has been recorded from two regions of Mauritius (Flacq and Plaines Wilhems). Likely, the location at Flacq does not exist anymore (C. Baider, 2018, personal communication). The species prefers mid to higher elevations on La Réunion, while it it is found at lower elevations on Mauritius. It grows in ericoid thickets, forests, on boulders or in clumps in seasonally-flooded upland marshes. Its area of occupancy was estimated as 64 km2 and its EOO as 6,805 km2, and it occurs at four locations (the Parc National de La Réunion, and in the Black River Gorges National Park, the Perrier Nature Reserve and the protected areas of the Bambou Mountains on Mauritius). The habitat of the species in Le Réunion is threatened by invasive alien species, disturbance due to human activities, and climate change. In Mauritius, similar threats to the habitat of the species exist, in particular due to invasive alien species and the patchiness of the remaining native vegetation. Therefore, Costularia melicoides is here assessed as Endangered EN B2ab(i,ii,iii,iv).

Notes

In Costularia melicoides, the lower fertile flower is bisexual and the upper fertile flower is male (or sterile), in contrast with the mainland African and Madagascan Costularia species (with lower fertile flower male or sterile, and upper fertile flower bisexual), and in contrast with Costularia cadetii from La Réunion with two bisexual flowers.

Notes

All specimens included in the molecular phylogenetic study were collected in La Réunion.

9. Costularia melleri (Baker) C.B.Clarke ex Cherm., Cat. Pl. Madag., Cyper. 40. 1931 (Costularia melleri C.B.Clarke, Consp. Fl. Afr. 5: 658. 1894, nom. inval.) ≡ Cladium melleri Baker, J. Linn. Soc., Bot. 21: 451. 1885 ≡ Mariscus melleri (Baker) Fernald, Rhodora 25: 54. 1923 ≡ Machaerina melleri (Baker) T.Koyama, Bot. Mag. (Tokyo) 69: 64. 1956 ≡ T. melleri (Baker) T.Koyama, J. Fac. Sci. Univ. Tokyo, Sect. 3, Bot. 8: 75. 1961—Fig. 2

Type (lectotype designated here). Madagascar, Antananarivo, between Toamasina and Antananarivo, July–August 1862, C.J. Meller s.n. (lectotype: K000244888!, isolectotype: MNHN-P-P00459987!).

Perennial herb up to 180 cm tall with a short, woody rhizome. Culm 80–100 cm × 4–9 mm, robust, slightly compressed, smooth-grooved, tapering to the top. Basal leaves leathery; leaf blades 30–36 cm × 7–15 mm, flat or with inrolled edges, margins scabrid, tapering strongly above the leaf sheaths, very acute; leaf sheaths much broader, 8–10 × 3.5 cm, dark brown-purplish. Cauline leaves 3, up to c. 35 cm, widely spaced; sheaths scarcely enlarged brownish-green base brown, edge obliquely cut. Inflorescence a large panicle, 60–100 cm long, up to c. five to seven cm wide; inflorescence bracts 9–11, sheathing, dark brown-purple. Peduncles unequal, up to 12 cm long, quite robust, flattened, margins scabrid. Pedicels of the spikelets three to five mm long, flattened, margins scabrid, ± curved. Spikelets very numerous, linear-oblong, 7–10 × 1–1.5 mm. Glumes 16–18, pale reddish or light reddish brown with hyaline margins, obtuse, lower 14–16 empty, upper 2 fertile; lower empty glumes very small, increasing in size towards top of spikelet; top glume somewhat reduced, pale, narrow. Flowers 2, lower male or sterile, upper bisexual. Perianth bristles 6, two to three times as long as the nutlet including its beak, rust-coloured, long ciliate, plumose. Stamens 3; anthers linear, shortly apiculate. Style trifid, long, hispidulous, pale brown, base pyramidal or triangular persistent. Nutlet obovoid, quite strongly trigonous, with canaliculate ribs, 1.5 mm long, rugulose, reddish brown; beak one mm long.

Distribution

Endemic to Madagascar, occurring in the provinces Antananarivo, Fianarantsoa and Toamasina (Fig. 2).

Ecology

Marshes, humid areas in forest, an elevation of 1,000–1,500 m.

Phenology

Specimens with very young inflorescences were found in October, November and April, flowering specimens were collected in November and January, fruiting specimens in December. Specimens collected in March and April had already lost their ripe nutlets.

Conservation status

The conservation status of Costularia melleri was previously assessed by Faranirina (2017) as EN B2ab(i,ii,iii,iv,v) based on an estimated AOO of 45 km2 (within the limits for EN status under the criterion B2) and five known locations. Only one subpopulation occurs in a protected area (Ranomafana National Park; Larridon et al. 2010-0249), the other subpopulations are known from unprotected areas subject to agriculture activity (Faranirina, 2017). Faranirina (2017) projected that the ongoing loss of its habitat will induce a strong continuous decline in the number of subpopulations and mature individuals in the next ten years as well as a continuing decline in its EOO and AOO.

Notes

Several specimens (i.e. Baron 1026, Baron 4104, Bosser 122, Decary 5826, Du Petit Thouars s.n. and Meller s.n.) have fewer glumes but represent very young plants. Clarke (1894) did not validly publish the combination Costularia melleri. He stated it to be a ‘sp. nov’., although he cited the three syntypes of Cladium melleri Baker (Baron 1026, Baron 2846 and Meller s.n.) and a 4th specimen (Baron 4104). The name lacks any reference to the basionym ICN Art 41.1 and lacks any form of description, so this does not constitute valid publication of the combination according to Art. 41.4 (Turland et al., 2018).

10. Costularia microcarpa (Cherm.) Kük., Repert. Spec. Nov. Regni Veg. 46: 69 (1939) ≡ Costularia baronii var. microcarpa Cherm., Bull. Soc. Bot. France 72: 617 (1925)—Fig. 5

Type (lectotype designated here). Madagascar, Fianarantsoa, Isalo, 1,000 m, October 1924, H. Perrier de la Bâthie 16704 (lectotype: MNHN-P-P00723561!, isolectotypes: MNHN-P-P00459969!, MNHN-P-P00459970!, MNHN-P-P00459971!).

Very robust and tall perennial herb: Culm robust, 1.2–2 m × c. 6 mm. Basal leaves firm, long, five mm wide, flat, edges denticulate, involute, long attenuated; leaf sheaths up to 9 cm × 8–10 mm, brownish. Inflorescence a long, dense and complex panicle, built up from multiple branched fascicles; inflorescence bracts setaceous, shorter than fascicles, sheaths long and brown. Peduncles longest c. 7.5 cm. Pedicels of the spikelets two to four mm long, curved. Spikelets very numerous, most individually pedicellate, less often sessile, oblong ± 6–7.5 × 1–1.5 mm, subterete, arcuate. Glumes subdistichous, coriaceous, straw-coloured to purplish-striate; lower 6–10 glumes empty, ovate, subobtuse, with sparsely ciliolate margins; upper 2 fertile glumes lanceolate, acute, the topmost glume somewhat reduced. Flowers 2, lower male, upper bisexual. Perianth bristles 6, overtopping the nutlet, plumose from base to tip. Stamens 3. Style long, trifid, base thickened, hispidulous apex persistent. Nutlet obovate swollen-trigonous, two mm long, base attenuate, reddish, slightly rugulose.

Distribution

Endemic to the Ihorombe region of Fianarantsoa province in Madagascar (Fig. 5).

Ecology

Found growing on shaded, humid sandstone in Isalo National Park at an elevation of c. 1,000 m.

Phenology

Only known specimen was collected in October as flowering.

Conservation status

Costularia microcarpa is endemic to Madagascar. It is only found in Ihorombe region of Fianarantsoa province at elevation 1,000 m in Isalo National Park. There is insufficient information available to assess the conservation status of this species since it is only known from its type specimen. Therefore, it is categorised as DD. Research is needed to investigate whether the population of this species at the only known location in Isalo National Park is still present.

Notes

This is one of only two Costularia species that could not be sampled for this study, as it is only known form the type specimen. Although unsure at this time, this species is likely part of the Costularia pantopoda species complex. Chermezon (1937) originally published this as a variety under Costularia pantopoda subsp. baronii (as Costularia baronii var. microcarpa) though Kükenthal (1939) later recognised this taxon at species level.

11. Costularia natalensis C.B.Clarke, Consp. Fl. Afr. 5: 658 (1894)—Figs. 15, 16

Figure 15 Illustration of Costularia natalensis (Hilliard & Burtt 14504 NU).

(A) Habit; (B) spikelet; (C) terminal fertile glumes bearing a bisexual and a male flower, note six feathery bristles, three filaments, anthers already shed. Scale bars: A = four cm, B–C = two mm. Illustration drawn by Jane Browning.

Figure 16 Distribution map of Costularia natalensis in southeastern Africa.

The distribution of the species was mapped using SimpleMappr.

Type (lectotype designated here). South Africa, KwaZulu-Natal, (without stated locality but probably Noodsberg (Burtt, 1988; Browning & Gordon-Gray, 1996)), J. Buchanan 152 (lectotype: K000244893!).

Adapted from Browning & Gordon-Gray (1996): Perennial herb up to 2.5 m tall, tufted; rhizome 1–1.5 mm in diameter, woody, erect, clothed in thick adventitious roots. Culm erect, 50–150 cm tall including inflorescence, 2.5–4.5 mm wide. Basal leaves spirodistichous; leaf sheaths persistent, up to 15 mm wide; leaf blades 30–60 cm × (1.5–)3–4 mm, gradually tapering to elongate curling apices ±1 mm wide, tough, glabrous, margins scabrous. Cauline leaves 2–4. Inflorescence a panicle of closely packed, erect spikelets grouped in ± elongated irregular clusters or appearing interrupted with ± nodding clusters, 55–95 cm long; inflorescence bracts 4–8(–12), dark chestnut brown to blackish-red, reducing in size upwards. Peduncles unequal, up to c. 16 cm long, flattened, scabrid on the margins. Pedicels of the spikelets 2.5–6 mm long, straight to curved, very scabrid. Spikelets oblong, 6–9 × 1.8–2.0 mm, dull dark brown. Glumes subdistichous, 6–12, lower three to nine empty, of which lowest one to three frequently with apex attenuate, remainder increasing in length upwards, apices acuminate or acute, next three largest, 6–7 × 3 mm, boat-shaped, glabrous except for well-marked ciliate margin, apex obtuse, toothed, but rolled so appearing narrow, and almost acute until unfolded, uppermost glume enclosed within the two preceding, slightly shorter. Flowers 2, lower male, upper bisexual. Perianth bristles 6, delicate, six to seven mm long, white, villous in distal half. Stamens 3, filaments persistent five to seven mm long after anthesis, ribbon-like; anthers linear-oblong, large, apiculate, early deciduous. Style trifid, dark brown, coarsely plumose, proximal portion of style persistent as short to long beak on fruit. Nutlet rounded trigonous, narrowed basally into funnel-shaped extension ±1/4 length of globose portion, 5 × 3 mm in total length and width, faintly 3-ridged longitudinally, whitish to pale fawn; surface smooth to slightly transversely rugose.

Distribution

Costularia natalensis is restricted in its distribution to southeastern Africa (Fig. 16). In particular, the species is present at higher elevation (1,070–2,130 m) along the chain of individually isolated highlands roughly paralleling part of the coastline, for example, in South Africa the Wolkberg, Sabie and Graskop areas of the Mpumalanga Drakensberg (Browning & Gordon-Gray, 1996). Mount Mulanje in Malawi is the northernmost known locality, and the southernmost distribution of the species reaches the area of Pietermaritzburg in Kwazulu-Natal (South Africa).

Ecology

According to observations by Browning & Gordon-Gray (1996), populations are mostly very localised, often small, and in KwaZulu-Natal, frequently limited to a few scattered, solitary plants which grow on steep, rocky slopes, associated with coarse grasses in the zone between forest and grassland. A slightly more extensive population grows along banks of small streams and among boulders, where nutrients particularly phosphates are in short supply and other vegetation is scare (Restionaceae and short grasses), in the Chimanimani National Park (Zimbabwe; Browning & Gordon-Gray, 1996). In Mozambique, several (small) subpopulations are also found on quartzite sandstone in the Chimanimani Mts, and on rocks in the submontane grasslands of Mt Gorongosa and Serra Choa. Plants of this species have been collected from Mount Mulanje in Malawi (which is composed of seynite, quartz-seyinite and granite rock materials), in particular from the eastern zone of the Biosphere Reserve (Lichenya and Chambe). It is restricted to higher elevations.

Phenology

Flowering/fruiting specimens were collected from November to May.

Conservation status

Costularia natalensis is restricted in its distribution to southeastern Africa (Malawi, Mozambique, Zimbabwe, Swaziland, South Africa). It is found at higher elevations in rocky areas in grassland and shrubland. Threats affecting part of the range of the species include fire, fuelwood collection, illegal logging of natural forests and plantation forestry, invasive species and potential mining. Although the population of this species is believed to be decreasing (Browning & Gordon-Gray, 1996), it currently does not fall within the criteria for any of the threat categories, and is therefore assessed as LC. However, further research is needed to investigate threats and population size.

Notes

In our molecular phylogenetic results (Fig. 1), Costularia natalensis is found in a clade with three other Costularia species, two of which are here described as new to science, that is, Costularia andringitrensis and Costularia itremoensis. Although Burtt (1988) indicated some morphological variety between plants of different localities, Browning & Gordon-Gray (1996), who studied specimens from the entire distribution range of Costularia natalensis, found no clear discontinuities that may provide a basis for subdivision of the species.

12. Costularia pantopoda (Baker) C.B.Clarke ex Cherm., Arch. Bot. Bull. Mens. 7 (Mém. 2): 80. 1936 ≡ Cladium pantopodum Baker, J. Linn. Soc., Bot. 21: 451. 1885 ≡ Mariscus pantopodus (Baker) Fernald, Rhodora 25: 54. 1923 ≡ Machaerina pantopoda (Baker) T.Koyama, Bot. Mag. (Tokyo) 69: 65. 1956 ≡ T. pantopoda (Baker) T.Koyama, J. Fac. Sci. Univ. Tokyo, Sect. 3, Bot. 8: 75. 1961.

Type (implicitly lectotypified byClarke (1894): 658). Madagascar, Central Madagascar, R. Baron 2072 (lectotype: K000244886!, isolectotypes: K001322342!, K!, MNHN-P-P00459988!).

Conservation status

Costularia pantopoda is restricted in its distribution to south-central Madagascar, and occurs in at least one protected area, that is, the Andringitra National Park. Threats to this taxon need further investigation but in the Andrigitra National Park, where most collections have been made, its habitat and area of occupancy are impacted negatively by cattle grazing and by fires started for pastoral reasons which can easily get out of control and enter the National Park (I. Larridon, 2010, personal observation; F. Rakotonasolo, 2017, personal observation). In other areas, fire (natural and man-made) and disturbance of its habitat as a result of logging, firewood collection and charcoal may also affect this species. Based on 10 georeferenced herbarium specimens, the taxon occurs in at least six locations and has an estimated AOO of 40 km2 and an EOO of 9,478 km2. Using IUCN criteria, this variety can be assessed as VU B1ab(ii,iii)+2ab(ii,iii).

12a. Costularia pantopoda (Baker) C.B.Clarke ex Cherm. var. pantopoda—Figs. 5, 17, 18

Figure 17 Illustration of Costularia pantopoda var. pantopoda (Larridon et al. 2010-0144 GENT).

(A) Habit; (B) inflorescence matching the habit; (C) spikelet; (D) lowest glumes of spikelets; (E) rest of glumes lower to upper, abaxial view; (F) bisexual flower; (G) male flower; (H) nutlet. Scale bars: A–B = three cm, C–H = three mm. Illustration drawn by Juliet Beentje.

Figure 18 Habitat and morphology of Costularia pantopoda var. pantopoda.

(A) Plant in situ; (B) inflorescence detail. Photos taken by Muthama Muasya in Adringitra National Park, Madagascar on 18 April 2010.

Medium-sized perennial herb, up to c. 65 cm height. Culm 25–30 cm × 2.5–3 mm. Basal leaves distichous; leaf sheaths 6–7 (–9) × c. 2 cm, brownish-purple, very wide compared to the leaf blades; leaf blades usually enrolled and thus appearing much narrower than the leaf sheaths, one to four mm wide when enrolled, up to c. seven mm when flattened, leathery, margins scabrid. Cauline leaves 1–2, five to seven mm wide margins scabrid, sheaths brownish. Inflorescence a panicle 10–35 cm long, quite tight, narrow; inflorescence bracts 5–10, brown to dark purple. Peduncles longest four to five cm, erect to arching downwards, flattened, margins scabrid. Pedicels of the spikelets erect, two to six mm, margins scabrid. Spikelets lanceolate, 6–7.5 × 1.5–2.3 mm. Glumes 5–8, the largest 5–6.5 mm long, dark purplish-brown to nearly black, the lower three to six empty, ovate, scabrid on the keel, ciliolate at the apex, increasing in length; two upper flowering glumes ovate-lanceolate. Flowers 2, lower male, upper bisexual. Perianth bristles 6, much longer than the nutlet, plumose, long ciliate. Stamens 3; anthers linear, shortly apiculate. Style trifid, long; style base hispidulous, triangular, persistent. Nutlet broad obovoid triangular, somewhat rugulose, 1.75 mm; beak one mm, almost as wide as the nutlet.

Distribution

Costularia pantopoda var. pantopoda occurs in the Fianarantsoa province and in the south of the Antananarivo province of Madagascar (Fig. 5).

Ecology

Rocky areas at high elevation (1,300–2,500 m).

Phenology

Flowering specimens were collected from December to April, fruiting specimens from September to November.

Notes

Baker (1885: 451) originally described Cladium pantopodum based on two specimens collected by Baron (2072 and 3316). In 1894, Clarke placed this species in Costularia and split it up into two species, that is, Costularia pantopoda (Baron 2072) and Costularia baronii (Baron 3316, Baron 4517, Baron 5061, Scott Elliot 1989). However, Clarke (1894: 658) failed to provide a description for Costularia baronii, this species was only made valid in Clarke (1897–1898: 274) where he provides a short diagnosis for it at the end of his treatment of Costularia natalensis.

12b. Costularia pantopoda var. gracilescens Kük., Repert. Spec. Nov. Regni Veg. 41: 67 (1939)—Fig. 6

Type (lectotype designated here). Madagascar, Antananarivo, Antsirabe, 1,600 m, January 1919, H. Perrier de la Bâthie 2729 (lectotype: P; isolectotype: K000244887!).

Culms slender, 0.9 mm wide. Basal leaves with the leaf sheath little larger than the narrow leaf blades (up to c. 1.5 mm wide). Inflorescence fairly contracted panicle, with fewer spikelets, and composed of four to five fascicles. Peduncles longest 8.5 cm. Pedicels of the spikelets two to eight mm long, flattened, margins scabrid. Spikelets 6–7 × 1.5 mm, purple. Glumes clearly distichous, lower five to six empty glumes mucronate to acute, upper two fertile glumes obtuse. Perianth bristles 6, 5.5 mm long, pale to rusty-coloured, antrorsely ciliate. Stamens 3; anthers 3.5 mm, linear, apiculate. Nutlet immature.

Distribution

Known from a single collection made near Antsirabe in the Antananarivo province of Madagascar (Fig. 5).

Ecology

The only known collection was found growing in a marsh at c. 1,600 m in elevation.

Phenology

The taxon was collected in flower in January.

Notes

Kükenthal (1939) described this new variety based on a single specimen (Perrier de la Bâthie 2729). Though likely present in P, the lectotype could not be traced. This variety most closely resembles Costularia pantopoda var. pantopoda.

12c. Costularia cf. pantopoda—Figs. 5, 19

Figure 19 Illustration of Costularia cf. pantopoda (Humbert 30061 P).

(A) Habit; (B) inflorescence matching the habit; (C) detail of inflorescence; (D) spikelet; (E) bisexual flower and glume; (F) male flower and glume. Scale bars: A–B = three cm, C = one cm, D–F = three mm. Illustration drawn by Juliet Beentje.

Specimen. Madagascar, Toamasina, Ambatofinandrahana-Amborompotsy, Mountains W of Itremo (W Betsileo), 1,500–1,700 m, 17–22 January & 18–22 April 1955, H. Humbert 30061 (MNHN-P-P01908604!).

Notes

A single specimen was collected from the mountains West of Itremo, at an elevation of 1,500–1,700 m outside of the Itremo protected area delimitation. Grazing, fire (natural and man made) to renew cattle pasture and mining are the major threats wich affect this habitat. This specimen appears as sister to Costularia pantopoda in the phylogenetic hypothesis (Fig. 1). This specimen shows some similarities with Costularia itremoensis (Figs. 12, 19).

13. Costularia purpurea Cherm., Bull. Soc. Bot. France 69: 722. 1922 publ. 1923 ≡ T. purpurea (Cherm.) T.Koyama, J. Fac. Sci. Univ. Tokyo, Sect. 3, Bot. 8: 75. 1961—Fig. 14

Type (lectotype designate here). Madagascar, Antananarivo, Analamanga, Manjakandriana, forest E of Ambatolaona, 1,300–1,450 m, 11 November 1912, R. Viguier & H. Humbert 1231 (lectotype: MNHN-P-P00459990!; isolectotypes: MNHN-P-P00459991!, MNHN-P-P00459992!).

= Costularia laxa Cherm., Bull. Soc. Bot. France 69: 723. 1922 publ. 1923 ≡ T. laxa (Cherm.) T.Koyama, J. Fac. Sci. Univ. Tokyo, Sect. 3, Bot. 8: 75. 1961. Type (lectotype designated here). Madagascar, Antsiranana, Manongarivo, 1,000 m, May 1909, H. Perrier de la Bâthie 2639 (lectotype: MNHN-P-P00459983!; isolectotype: MNHN-P-P00459984!).

Perennial herb up to 2.5 m tall. Caudex covered in lateral roots can be present (0.7–1.5 cm in diam.). Culms more or less robust, 50–1.3 m × 2.5–4 mm. Basal leaves distichous; leaf sheaths 3.5–8.5 cm × up to 9 mm, brownish-purple; leaf blades leathery, (28–)50–80 cm × 3–8 mm, flat or slightly inrolled, margins scabrid, tapering to a very acute tip. Cauline leaves 3–4, far apart; sheaths long tubular, purplish or brownish-green, mouth obliquely cut. Inflorescence a quite narrow panicle with lax partial inflorescences to a lax complex panicle, 50–90(–165) cm long; inflorescence bracts 8–14; sheaths purple. Peduncles unequal, up to 13 cm, flattened, margins scabrid. Pedicels of the spikelets 5–10(–25) mm flattened, margins scabrid, straight or curved. Spikelets oblong-lanceolate, very flattened, (4–)6–10 × (1.2–)2 mm. Glumes (dark) purple, ovate-lanceolate, (sub)acute, keel scabrid, edges minutely ciliate, (3–)5–9 lower glumes empty, two upper fertile, largest 4–5.5 mm long. Flowers (1–)2, either both bisexual (generally only lower perfecting a nutlet), or lower bisexual and upper functionally male, more rarely lower male and upper bisexual, or rarely a single bisexual flower. Perianth bristles 6, equalling or surpassing the nutlet, pale reddish-brown, plumose below, densely and shortly ciliolate above. Stamens 3; anthers linear, reddish, connective very shortly apiculate. Style long, deeply trifid, thin, brownish; style base triangular, hispidulous, pale, persistent. Nutlet rounded-trigonous, (1.5–)2–3 mm × 1–1.5 mm, greyish-brown, rugulose- reticulate, with an attenuate base; beak (1–)1.5–2.5 mm long, base as wide as the nutlet.

Distribution

Costularia purpurea is endemic to Madagascar, occurring in the Antananarivo, Antsiranana, Fianarantsoa, Toamasina and Toliara provinces (Fig. 14).

Ecology

The species is found infrequent on granitic formations in ericaceous shrubland, grassland and open forests at mid to high elevations (500–1,850 m).

Phenology

Flowering/fruiting specimens were collected from November to May. Young inflorescences can be observed on the specimens collected in September–October, while old inflorescences remain on the plants until September.

Conservation status

The species occurs in a range of protected areas including: Analamazaotra (Périnet), Andohahela, Didy National Park (NP), Kalambatritra, Manjakatompo Ankaratra, Manongarivo Reserve, Marojejy NP, Masoala NP, and Ranomafana NP. Based on its known and projected distribution, it is likely also present in among others Midongy du Sud NP. Since no specific threats are known to the species, and because it has a wide distribution in Madagascar (AOO = 132 km2, EOO = 218,948 km2) and occurs in a range of protected areas, Costularia purpurea is here assessed as LC.

Notes

The specimen Hildebrandt 3752a was listed as a syntype of Costularia recurva (accepted name Costularia leucocarpa) but conforms to circumscription of Costularia purpurea. A lot of confusion existed between Costularia leucocarpa and Costularia purpurea, with many Costularia purpurea specimens at the G, K and P herbaria identified as Costularia recurva. However, these species are quite different in morphology, with Costularia purpurea characterised by longer, flatter, narrower, darker spikelets generally bearing more glumes compared to Costularia leucocarpa.

Most herbarium specimens listed as Costularia laxa by Chermezon (1937) and Kükenthal (1939) are very immature and difficult to identify. Although the clade with two accessions originally identified as Costularia laxa and three accessions identified as Costularia purpurea is well supported in the molecular phylogenetic hypothesis, the taxa themselves are not (Fig. 1). Chermezon (1937) and Kükenthal (1939) distinguished Costularia laxa from Costularia purpurea based on it laxer inflorescence, fewer empty glumes (3–4 vs. 5–9) and perianth bristles much overtopping the nutlets. However, this distinction does not hold as variation in inflorescence branching and number of spikelets per inflorescence is gradual, even in the specimens listed by Chermezon (1937) as Costularia laxa spikelets often have more than four empty glumes, and in Perrier de la Bâthie 2639 (MNHN-P-P00459983), selected as hololectotype of Costularia laxa, the size of the nutlets varies from 1.6 to 2 mm plus a beak of 0.7–1.3 which is not that much shorter than the perianth bristles and similar to many specimens conforming to the description of Costularia purpurea.

This species is sister to Costularia melicoides of the Mascarenes. Costularia melicoides is unusual in perfecting a nutlet in lower of the two fertile glumes, while most Costularia species perfect a nutlet in the upper fertile glume. In Costularia purpurea, the number and sex of the flowers is variable with many of the collected specimens also perfecting a nutlet in the lower fertile glume. These sister species also share a similar build and size.

Costularia laxa var. macrantha Cherm. (1925: 21) is here excluded from Costularia laxa as we consider it to be synonymous with Costularia robusta (see more discussion under that taxon).

14. Costularia robusta (Cherm.) Larridon, comb. et stat. nov. ≡ Costularia baronii C.B.Clarke var. robusta Chermezon, Bull. Soc. Bot. France 69: 723 (1922) ≡ Costularia pantopoda var. robusta (Cherm.) Kük., Repert. Spec. Nov. Regni Veg. 41: 68 (1939)—Fig. 5

Type. Madagascar. Antisiranana, Diana, (Tsaratanana Reserve, Maromokotro), 2,700 m, December 1912, H. Perrier de la Bâthie 2503 (holotype: MNHN-P-P00459966!).

= Costularia laxa var. macrantha Cherm., Bull. Soc. Bot. France 72: 21. 1925. Type (lectotype designated here). MADAGASCAR, Antsiranana, Diana, (Tsaratanana Reserve, Maromokotro), 2,000 m, Janaury 1923, H. Perrier de la Bâthie 15652 (lectotype: MNHN-P-P00459967!; isolectotype: MNHN-P-P00459968!).

Very robust and tall perennial herb with a strongly developed and long (c. one m) caudex. Culms 0.8–2 m × c. 6 mm, robust. Basal leaves with very wide leaf sheaths (15–20 mm), persistent at the base of the culm above the caudex. Inflorescence an elongate, narrow panicle with very numerous, crowded spikelets; inflorescence brances erect, not more than five cm long. Pedicels of the spikelets short, not patent. Spikelets six to seven mm long. Glumes 12–14, up to seven mm long, purplish black.

Distribution

Costularia robusta is only known from the Manongarivo, Marojejy and Tsaratanana protected areas and their environs in the Antsiranana province of Madagascar (Fig. 5).

Ecology

This taxon occurs in ericoid shrublands at (very) high elevations (1,400–2,800 m).

Phenology

Inflorescences are initiated in April and flower/fruit between October and January. Old inflorescences are still visible on the plants in April when the new inflorescence are formed.

Conservation

Costularia robusta is restricted in its distribution to the Antsiranana province of Madagascar, and occurs in at least three protected areas, that is, Manongarivo, Marojejy and Tsaratanana. Threats to this taxon need further investigation but fire (natural and man-made) and disturbance of its habitat as a result of logging, firewood collection and charcoal may affect this species. Based on 11 georeferenced herbarium specimens, this species occurs in at least six locations and has an estimated AOO of 28 km2 and an EOO of 2,947 km2. Using IUCN criteria, Costularia robusta can be assessed as VU B1ab(ii,iii)+2ab(ii,iii).

Notes

Kükenthal (1939) also listed the specimen Humbert 3344 when creating the combination under Costularia pantopoda. We here exclude this specimen collected at Pic d’Ivohibe Reserve in the Fianarantsoa region from Costularia robusta and place it in Costularia baronii. Kükenthal (1939) listed specimen Perrier de la Bâthie 16398 under Costularia pantopoda var. baronii. Although this specimen could not be found in the P herbarium, it is very likely to be Costularia robusta as it is from the same locality as the other positively identified specimens of Costularia robusta by the same collector at the same time.

Costularia laxa var. macrantha looks very similar to Costularia robusta but has paler glumes. Since its type specimen was collected at a somewhat lower elevation than the specimens identified as Costularia robusta, this is unsurprising since glume colour in tropical Cyperaceae often darkens with elevation in the same species (I. Larridon, 2010, personal observation). Kükenthal (1939) discusses the presence of a caudex in Costularia robusta. Although this is not clearly visible in the specimens he cited, the type specimen of Costularia laxa var. macrantha clearly has a well developed and long caudex. We here consider Costularia laxa var. macrantha to be a synonym of Costularia robusta. Kükenthal (1939) did realise a potential relationship between Costularia laxa var. macrantha and Costularia pantopoda as he included a comment under Costularia laxa var. macrantha to the effect of ‘Much deviating from the typical form of Costularia laxa and approaching Costularia pantopoda in appearance, but the glumes have the colour of Costularia laxa (purple and pale green) and the perianth bristles are more ciliate than plumose. Possibly, a hybrid between Costularia laxa and Costularia pantopoda’. Kükenthal (1939) listed a second specimen under Costularia laxa var. macrantha: H. Humbert 6358 (not seen) collected in Beampingaratra, Toliara province. We here exclude this specimen from Costularia robusta.

15. Costularia xipholepis (Baker) Henriette & Senterre, Phytotaxa 231: 34 (2015). ≡ Cladium xipholepis Baker, Fl. Mauritius: 424 (1877). ≡ S. xipholepis (Baker) Summerh., Bull. Misc. Inform. Kew 1928: 394 (1928), p.p. quoad holotypus sed excl. Horne 626—Fig. 20

Figure 20 Distribution map of Costularia xipholepis in the Seychelles (Mahé island).

The distribution of the species was mapped using SimpleMappr.

Type. Seychelles, Wright s.n. (holotype K!).

Adapted from Henriette et al. (2015): Perennial herb up to 2.5 m tall, caespitose, forming dense clumps. Culm c. 80 cm × 3.5–5 mm, cylindrical, robust. Basal leaves distichously arranged, densely set, numerous; dead leaves persistent, the older ones abscising above the leaf sheath; green leaves 7–12 on each side, arcuate; leaf-sheath 4–6 × 2.9–4 cm, semi-cylindrical, thick, yellowish, margins dark red, ciliate distally; leaf blade 75–123 cm × 7–10 mm, not pseudopetiolate, linear, gradually tapering towards apex, upwardly concave in section, coriaceous, glabrous, smooth, mid-green, margin entire, with tiny ascendant prickles, apex acute, slightly rounded, not apiculate, midrib not distinct, longitudinally striate. Cauline leaves 3–5; leaf-sheath 4.5–5.2 × 1.2–1.5 cm, closed, dark red at base, yellowish distally; leaf-blade shorter than in basal leaves, decreasing in length towards the apex of the culm, 40–74 cm × 8–10 mm. Inflorescence 55–140 cm, narrow (7–15 cm wide), with four to five orders of branching; inflorescence bracts 9–14, up to 17–27 cm long at basal nodes, 2.5–3.0 cm long at distal nodes. Peduncles unequal (longer in basal fertile nodes), 15–360 mm long, one to seven per node, erect, compressed, slender, smooth. Pedicels of the spikelets 7.5–8.0 mm long, straight. Spikelets densely clustered, 7–8 × 1.0–1.2 mm, lanceolate, reddish-brown; rachilla persistent, straight. Glumes 7–9, distichous, completely enclosing the rachilla at base, deciduous, lanceolate, smooth, reddish-brown on the sides and towards apex, margins glabrous, apices with a straight awn (longer in basal glume), midrib distinct; basal empty glumes 5–7, the lowest glume 2.5–3.7 × 1.0–1.5 mm, subsequent glumes 3.3–6.8 × 1.4–2.0 mm; lower fertile glume 6.0–6.5 mm long, slightly shorter than the last empty glume; upper fertile glume 6.1–6.5 mm long, enclosed in the previous glume. Flowers 2, lower male, upper bisexual. Perianth bristles 6, well developed, 5.5–7.3 mm long, two to three times longer than the nutlet (beak included), sparsely plumose. Stamens 3, 5.0–7.6 mm long, not protruding or slightly protruding; anthers oblong, 1.7–4.2 mm long, yellow. Style trifid, 7.5–10.3 mm long (including stigmas). Nutlet stalked at maturity, trigonous, wingless, obovoid, two mm long (excluding beak), 0.8–0.9 mm diam., golden brown, beak with a constriction at the junction with the nutlet, 1.5 mm long, long-acute, 0.4 mm wide at base, ciliate.

Distribution

Based on Henriette et al. (2015), Costularia xipholepis is endemic to the Seychelles and restricted to Mahé and has been found in three locations all situated in the Morne Seychellois National Park: Congo Rouge (B. Senterre & T. Stévart, observation record, 20 July 2014, 4.6512°S, 55.44126°E, 610 m), Copolia and Pérard (Fig. 20). Two additional locations were recently discovered at Mont Sébert and at Glacis Sarcelles (B. Senterre, 2018, personal communication) (Fig. 20).

Ecology

This species is restricted to the herbaceous fringe of lower montane inselbergs (Henriette et al., 2015). It has been observed from 500 to 821 m but was more abundant on the site at the higher elevation, which corresponds to an altitudinal belt named the tree fern lower montane belt (Senterre, 2011; Senterre & Wagner, 2014; Senterre et al., 2009; Henriette et al., 2015). At Copolia, it has a patchier distribution, growing on rock crevices and along fissures where the soil is damp. In all sites, it grows in association with the species previously known as Costularia hornei (basionym S. hornei, nom. cons. prop.; Larridon, Govaerts & Goetghebeur, 2017a); which is now placed in the new genus Xyroschoenus (Larridon et al., 2018a). Since the exclusion of X. hornei from Costularia, only one species of Costularia is known to occur on the Seychelles.

Phenology

Flowering/fruiting specimens were collected between March and December.

Conservation status

Following Henriette et al. (2015), Costularia xipholepis is rare and highly localised. Three sub-populations representing three locations, 1.4–2.4 km apart, separated from each other by unsuitable habitat (i.e. wet forests) are within the Morne Seychellois National Park and appear healthy, with limited risks from invasive species. The AOO for the Congo Rouge population is 10 m2, Copolia 6,000 m2, and Pérard 20,000 m2 (Henriette et al., 2015). The two newly discovered sub-populations, at Mont Sébert and at Glacis Sarcelles are not well known, but the Mont Sébert one is about the same size as the one of Congo Rouge (small), while the Glacis Sarcelles population is comparable to the Copolia one (B. Senterre, 2018, personal communication) resulting in an estimated AOO totalling approx. 0.032 km2. Its EOO was estimated at approximately 5.9 km2 (Bachman et al., 2011). Both AOO and EOO fall within the limits of CR status under criterion B. According to Henriette et al. (2015) and based on IUCN (2012) criterion B, with an EOO < 5,000 km2, AOO < 500 km2, number of locations ≤5, and a projected decline of the quality of the habitat as a result of climate change, this species can be classified as Endangered EN B1ab(iii)+2ab(iii).

Notes

Henriette et al. (2015) noted that among the known species of Costularia s.s. only one presents some similarity with Costularia xipholepis, that is, Costularia baronii from Madagascar, as both species have long leaves and hypogynous bristles much longer than the nutlet with relatively few empty glumes. However, our molecular phylogenetic results point at a sister relationship with Costularia melleri. The sister pair Costularia melleri (Madagascar) and Costularia xipholepis (Seychelles) in turn are sister to a clade encompassing the species Costularia humbertii (Madagascar) and Costularia cadetii (La Réunion).

Conclusions

The genus Costularia is redelimited to represent a monophyletic entity including 15 species. Although the species diversity is largely found in Madagascar, our results indicate that the genus dispersed once to Africa (Malawi, Mozambique, South Africa, Swaziland, Zimbabwe), twice to the Mascarenes (La Réunion, Mauritius), and once to the Seychelles (Mahé). Three-quarters of the species are threatened with extinction, because of restricted distribution ranges and human impact. A full taxonomic treatment is provided, including an identification key to all species, formal descriptions of two new species from Madagascar (Costularia andringitrensis and Costularia itremoensis) and one new species from La Réunion (Costularia cadetii), and two taxa previously recognised as varieties of Costularia pantopoda are here recognised at species level (Costularia baronii and Costularia robusta).

Supplemental Information

Supplemental Information 1 List of the samples used in the molecular analysis with species names, voucher information, distribution and GenBank accession numbers for the DNA sequences of the three regions.

*indicates new accessions; a dash (–) indicates missing data.

Click here for additional data file.

Supplemental Information 2 Alignment of the DNA sequences of the three regions studied.

Click here for additional data file.

Supplemental Information 3 Additional herbarium specimens studied.

Click here for additional data file.

Supplemental Information 4 50% majority consensus single-locus BI tree (ETS) with the associated PP values.

Only posterior probabilities above 0.7 are shown.

Click here for additional data file.

Supplemental Information 5 Best scoring ML tree for ETS.

Only bootstrap values above 70% are shown.

Click here for additional data file.

Supplemental Information 6 50% majority consensus single-locus BI tree (ITS) with the associated PP values.

Only posterior probabilities above 0.7 are shown.

Click here for additional data file.

Supplemental Information 7 Best scoring ML tree for ITS.

Only bootstrap values above 70% are shown.

Click here for additional data file.

Supplemental Information 8 50% majority consensus single-locus BI tree (trnL-F) with the associated PP values.

Only posterior probabilities above 0.7 are shown.

Click here for additional data file.

Supplemental Information 9 Best scoring ML tree for trnL-F.

Only bootstrap values above 70% are shown.

Click here for additional data file.

Supplemental Information 10 Best scoring ML tree for the concatenated dataset.

Only bootstrap values above 70% are shown.

Click here for additional data file.

We thank the curators of the BR, G, GENT, K, L, MAU, P, TAN, and UPOS herbaria for the loan of specimens, imaging of specimens, and/or permission to carry out destructive sampling. We thank Pieter Asselman and Viki Vandomme for their help with the molecular lab work at Ghent University. The ANGAP Madagascar National Parks authority, the general secretariat of the AETFAT congress 2010 and the staff of the MBG office in Antananarivo are acknowledged for their help in securing collecting permits (N°082/10/MEF/SG/DGF/DCB.SAP/SLRSE–Isabel Larridon) for Cyperaceae in Madagascar and their help organising the expedition. We thank Vonona Randrianasolo (Kew Madagascar Conservation Centre), and Franck Rakotonasolo and Jacqueline Razanatsoa (Herbier du Parc Botanique et Zoologique de Tsimbazaza) for aiding with fieldwork. Thank you to Modesto Luceño and his student José Ignacio Márquez-Corro for collecting additional material of the new species from La Réunion for this study. Jeremy Bruhl, Wim Huygh and Modesto Luceño are acknowledged for providing photos of Costularia species. Special thanks to Juliet Beentje and Jane Browning for making line drawings of many of the species.

Additional Information and Declarations

Competing Interests

Author Contributions

Field Study Permissions

DNA Deposition

Data Availability

New Species Registration

The authors declare that they have no competing interests.

Isabel Larridon conceived and designed the experiments, performed the experiments, analyzed the data, contributed reagents/materials/analysis tools, prepared figures and/or tables, authored or reviewed drafts of the paper, approved the final draft.

Linah Rabarivola performed the experiments, analyzed the data, contributed reagents/materials/analysis tools, prepared figures and/or tables, authored or reviewed drafts of the paper, approved the final draft.

Martin Xanthos performed the experiments, authored or reviewed drafts of the paper, approved the final draft.

A. Muthama Muasya performed the experiments, contributed reagents/materials/analysis tools, prepared figures and/or tables, authored or reviewed drafts of the paper, approved the final draft.

The following information was supplied relating to field study approvals (i.e. approving body and any reference numbers):

Permits to collect and export these specimens were issued by the Madagascar authorities: a collecting permit for Cyperaceae in Madagascar (N°082/10/MEF/SG/DGF/DCB.SAP/SLRSE–Isabel Larridon) was provided by ANGAP Madagascar National Parks authority.

The following information was supplied regarding the deposition of DNA sequences:

The newly generated sequence data have been deposited to GenBank (accession numbers MH512812 to MH512851), and are available as part of Data S2 (sequence alignment used for the analyses).

The following information was supplied regarding data availability:

DNA sequence data has been submitted to GenBank (https://www.ncbi.nlm.nih.gov/genbank/). GenBank accession numbers are included in Table S1 (voucher information) and in Data S1 (sequence alignment used for the analyses).

Herbarium specimens studied are accessible in public herbaria (contact info via: http://sweetgum.nybg.org/science/ih/). The additional herbarium specimens studied per taxon are listed in Data S2.

The following information was supplied regarding the registration of a newly described species:

Costularia andringitrensis LSID: 77194632-1

Costularia cadetii LSID: 77194635-1

Costularia itremoensis LSID: 77194637-1

Costularia robusta LSID: 77194639-1.

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
