# Peer review of "Revision of the Afro-Madagascan genus Costularia (Schoeneae, Cyperaceae): infrageneric relationships and species delimitation"

_PeerJ, doi:10.7717/peerj.6528_

## Round 0.1 · original submission · Minor Revisions

Dear authors,

2 reviewers are quite positive with your submitted ms. Both suggest to minor changes, most of them in the annotated PDFs. Please consider these points for final acceptance of your ms.

Best wishes
Mike Thiv

Reviewer 1 ·

Basic reporting

The manuscript entitled “Revision of the Afro-Madagascan genus Costularia (Schoeneae, Cyperaceae): infrageneric reationships and species delimitation” is a quite complete and excellent revisión, using morphological and molecular tools, of Costularia, one of the least studied Cyperaceae genera until now.

English is correct and references are appropriate.

All figures and tables are an essential complement to understand the results and conclusions and the paper structure is also appropriate.

Introduction summarizes well the background of this taxonomically difficult group.

In most cases, morphological features match very well with molecular results and taxonomic conclusions are well supported.

I have indicate in the attached pdf file numerous corrections and comments that, in my opinión, could contribute to improve the manuscript. The most important weakness of the paper is relative to descriptions of the species, highly heterogeneous. Authors must systematize the lengh, order and content of them. Moreover, in several cases, monophyly of the species was not tested or scarcely supported, although I can understand that it is very difficult to get sufficient materials of several species and, anyway, I think that the taxonomic results would not be very different.

Experimental design

This article is suitable for publication in Peer Journal.

Conclusions are a very important contribution to the systematics of the family Cyperaceae.

Methodology is detailed and appropriate, although nowadays there are molecular tools more conclusives as the employ of Next Generation Sequencing; nevertheless, as it is a scarcely collected genus, I can understand the difficulties to use those techniques.

Validity of the findings

This paper shows important novelties for researchers in systematics, mainly for cyperologists.

The statistical data (support of the clades) are generally highly robusts.

Conclusions are coherent with results.

Few important speculations are made.

Additional comments

No comment

Annotated reviews are not available for download in order to protect the identity of reviewers who chose to remain anonymous.

·

Basic reporting

The present work presents a nice, clear and much needed revision of the problematic genus Costularia (Cyperaceae) in its entire distribution area.
According to the expertise in Cyperaceae of some of the authors, they present an accurate background of the topic, investigate the taxonomy of the group using accurate molecular and morphological approaches, and present the results and discussion in a perfectly clear manner. The English is professional and do not need correction, although I detected a few typos.
References, figures and tables are accurate (and very nice!). The new taxa they describe are prefectly justified. Even they contained the description of a clearly new species because of having too few material (although I would encourage the authors to do it in this paper).
Overall, it is a very nice piece of work, worth to be published in PeerJ.

Experimental design

The materials, tools, experimental approach and test designs are accurate for an extensive taxonomic study like the one that is presented here. The explanation of the methods is fine, although I suggest some minor improvements in the attached file. There is no doubt that the procedures have been performed with rigorous caution and ethics.

Validity of the findings

The authors present the taxonomic revision of a group that was needed of study. They present completely new data that accurately update with the use of modern tools other older works in the group. The work seems quite robust, their results sounds and the conclusions well stated.

Additional comments

I attach a commented file with additional suggestions that I consider that will improve the quality of the manuscript. Regarding the taxonomic treatments, the changes resquested in the first species must be performed over the entire set of synopses.

---

## Round 0.2 · accepted · Accept

Dear authors, Isabel,

Thank you for implementing minor changes. I think the ms. is in good order now and regard it as accepted. I have just made a very few changes in the newly uploaded text l. 77 & 484. You may check and approve them.

Best wishes

Mike

#